# Dietary Patterns and Interventions to Alleviate Chronic Pain

**DOI:** 10.3390/nu12092510

**Published:** 2020-08-19

**Authors:** Simona Dragan, Maria-Corina Șerban, Georgiana Damian, Florina Buleu, Mihaela Valcovici, Ruxandra Christodorescu

**Affiliations:** 1Department of Cardiology, “Victor Babeș” University of Medicine and Pharmacy, 2 Eftimie Murgu Square, 300041 Timișoara, Romania; simona.dragan@umft.ro (S.D.); buleu.florina@gmail.com (F.B.); mihaeladanielacardio@gmail.com (M.V.); ruxandra_christodorescu@yahoo.com (R.C.); 2Institute of Cardiovascular Diseases Timișoara, 13 Gheorghe Adam Street, 300310 Timișoara, Romania; 3Department of Functional Sciences, “Victor Babeș” University of Medicine and Pharmacy, 2 Eftimie Murgu Square, 300041 Timișoara, Romania

**Keywords:** pain, diet, nutrition, inflammation, gastrointestinal, chronic pain, supplements

## Abstract

Pain is one of the main problems for modern society and medicine, being the most common symptom described by almost all patients. When pain becomes chronic, the life of the patients is dramatically affected, being associated with significant emotional distress and/or functional disability. A complex biopsychosocial evaluation is necessary to better understand chronic pain, where good results can be obtained through interconnected biological, psychological, and social factors. The aim of this study was to find the most relevant articles existent in the PubMed database, one of the most comprehensive databases for medical literature, comprising dietary patterns to alleviate chronic pain. Through a combined search using the keywords “chronic pain” and “diet” limited to the last 10 years we obtained 272 results containing the types of diets used for chronic pain published in the PubMed database. Besides classical and alternative methods of treatment described in literature, it was observed that different diets are also a valid solution, due to many components with antioxidant and anti-inflammatory qualities capable to influence chronic pain and to improve the quality of life. Thirty-eight clinical studies and randomized controlled trials are analyzed, in an attempt to characterize present-day dietary patterns and interventions to alleviate chronic pain.

## 1. Introduction

Acute pain is determined by the presence of noxious stimuli and/or ongoing tissue damage immediately perceived by the conscious brain, while chronic pain (CP) usually lasts longer than three months, according to the most recent definition of the European Pain Federation or EFIC, part of the European Cancer Organization since 5 June 2020. This simple classification of 7 types of pain according to the causal factor includes both primary and secondary CP. Primary pain comprises fibromyalgia or nonspecific low-back pain and secondary pain is caused by an underlying disease: chronic cancer-related pain, chronic neuropathic pain, chronic visceral pain, chronic posttraumatic and postsurgical pain, chronic headache and orofacial pain, and chronic musculoskeletal pain [1].

Chronic cancer-related pain caused by the presence of cancer, associated metastases or after cancer treatments should be distinguished from the pain caused by other comorbid diseases. Pain management in cancer patients and survivors is an important clinical challenge because CP due to the cancer itself, its treatment and other causes may be concurrent [2].

The most common conditions for peripheral neuropathy include peripheral nerve injury and radiculopathy, painful polyneuropathy, trigeminal neuralgia, and post-herpetic neuralgia. Central neuropathic pain can be caused by spinal cord or brain injuries, multiple sclerosis, or it may be appear after a stroke [3].

Chronic visceral pain is secondary to a pathological condition arising from the internal organs of any anatomic region, from head to neck or thorax, abdomen to pelvis. The underlying mechanism may be represented by chronic inflammation, local vascular, or mechanical factors. Pain intensity does not fully correlate with the pathological process, and local conditions favor the persistence of pain even after interventions considered successful for causal treatment [4].

Chronic headache or orofacial pain is the pain lasting at least 2 h per day, three months consecutively [5]. Pain of the scalp, face, eyes, mouth, and nose may be the sign of a lot of diseases, including inflammatory and neuropathic syndromes [6].

Chronic musculoskeletal pain is generally associated with persistent proinflammatory states that determine chronic pain due to changes in neuronal structure and sensitization which alter nociception [7]. It originates in the musculoskeletal structures or is related to somatic lesions followed by structural or biomechanical local changes [2].

The April 2019 11th edition of International Classification of Diseases (ICD) for mortality and morbidity statistics uses slightly different criteria of recognition of CP as health condition. Pain is divided into three categories: chronic pain, acute pain, and pain from unspecified cause. The third category excludes headache disorders, abdominal or pelvic pain, breast pain, pain of the joints, pain of the eyes, chest pain, perineal pain, pain of the shoulders, spinal pain, tooth pain, renal pain, pain of the throat, low back pain and limb pain, remaining only the pain not referable to any given organ or body region [8].

The Global Burden of Disease Study 2016 concluded that due to their high prevalence, pain and pain-related diseases are the leading causes of disability worldwide [7]. The burden of chronic pain is constantly escalating, recurrent tension-type headaches being considered the most frequent cause of pain, which affects 1.9 billion people yearly. Low back and neck pains also occupy leading positions when measuring years lived with disability in the top 10 classification of chronic pain [9,10]. In Europe the population is divided in high and low risk groups for developing chronic pain. In developing countries there are fewer groups at risk for developing chronic pain, while in less developed countries, the tendency is towards increase of populations at risk [9,10]. Chronic pain determines social burden especially in the most vulnerable subgroups, i.e., elderly, unemployed, and less educated, therefore a special attention to also relieve psychosocial components of chronic pain is recommended [11].

An unhealthy diet, obesity, smoking, and stress have a negative impact in the management of CP [12]. The human body uses six categories of nutrients from food: carbohydrates, fat, protein, fiber, minerals, and vitamins. Diet therapy represents a professionally prescribed diet which provides specific nutrients, antioxidants, or prebiotic supplementation for beneficial health effects. Dietary patterns in chronic pain also seem to have a positive impact on comorbidities including obesity, type 2 diabetes mellitus (T2DM), cardiovascular diseases, and depression [13,14,15,16].

The Western diet based on processed meat, sugary foods, refined grains, and low intake of fruits and vegetables causes an excessive production of proinflammatory mediators that sensitize the peripheral afferent neurons including interleukins, histamine, TNF-α, 5-hydroxytryptamine, bradykinin, free radicals, and eicosanoids (prostaglandins, leukotrienes, and thromboxane). Western diet imbalance also yields fewer anti-inflammatory mediators, including antioxidants and antioxidant defense. The composition of a western-style diet may not necessarily increase inflammation directly but rather induce a reduction in anti-inflammatory defense [17,18,19]. Chronic pain often results from a persistent proinflammatory state [20]. The Dietary Inflammatory Index has been developed to classify a person’s diet in order to help reverse the dietary imbalance by proper individualized diet therapy [21]. Alleviation of chronic pain can be obtained by reducing the intake of proinflammatory foods and increasing the intake of unsaturated fats, fruits, and vegetables. The requirements of such diets are fulfilled by diets high in whole-grains, fish, fruits, green vegetables, and olive oil [22]. Recent studies have identified many dietary choices that can improve CP due to components with antioxidant and anti-inflammatory properties [23]. Anti-inflammatory effects of polyphenols, the most abundant antioxidants in diet, are associated in many reviews with reduced CP, such as diabetic neuropathy [24,25], rheumatoid diseases [26,27], or even chronic pancreatitis [28].

Besides inflammation, low dietary intake of micronutrients, especially omega-3 fatty acids, vitamins B1, B3, B6, B12 and D, magnesium, zinc and β-carotene, is also associated with chronic neuropathic or inflammatory pain [13]. Supplementation of diet with these specific nutrients contributes to alleviation of CP [29], as observed in systematic reviews on chronic pelvic pain [22], low back pain [30], rheumatoid arthritis or joint pain secondary to inflammatory bowel disease [31,32], migraine [33], chronic noncancer pain [34,35], pain in chronic liver disease [36], or aromatase inhibitor-related arthralgia in breast cancer [37].

Gut microbiota contributes to digestion and absorption of food and a normal immune function, while dysbiosis is associated with the irritable bowel syndrome and chronic abdominal pain [38,39]. Prebiotics and probiotics represent promising solutions for regulating this disorder [40,41].

Of all well-known old and new lifestyle factors, dietary choices might influence the most the occurrence, maintenance, and perception of CP [15]. However, the effects of diet therapy in chronic pain have been analyzed in systematic reviews and meta-analyses and have provided ambiguous results [34,42]. The aim of this study was to find and analyze the most important original articles or reviews existent in the PubMed database, referring to dietary patterns and interventions used to alleviate chronic pain that dominated the last decade, in an effort to synthetize and classify current information and to draw conclusions on future directions of research.

## 2. Materials and Methods

The searches included, as a first step, only the keywords “chronic pain” in the PubMed database, one of the most comprehensive databases in medical literature. Then we have added the following keyword “diet”. Until 20 June 2020, the results of the searches in the PubMed database using as keyword “pain” [All Fields] generated 382,431 articles, published in the last 10 years. After restricting the search to “chronic pain” [All Fields] 66,593 results remained. The combined search using both keywords “chronic pain” and “diet” left 487 results in humans. We performed advanced search for each pain category and “diet” or “dietary intervention“. We refined the search using the NCBI filters: clinical trial, meta-analysis, randomized controlled trial, review and systematic review in the last 10 years in humans, leaving 272 results, out of which 213 were reviews and 59 original articles. After careful examination of the articles and elimination of 2 case-reports, 10 other articles referring in fact to acute post-surgical pain, 7 general lifestyle interventions without specific reference to the diets used and 2 supplementation diets that were not related to chronic pain, only 38 clinical trials, pilots and RCTs (Randomized Control Trials) remained. We also eliminated from our analysis 141 reviews that were either incomplete in data inclusion or comments or did not offer enough information to allow a proper comparison with the results found in our selection (Figure 1).

## 3. Results

Our search found 38 clinical studies, pilots, and randomized controlled trials made on types of diets reported to alleviate symptoms in various situations of chronic pain in the last 10 years (Table 1). Dietary patterns and interventions found in the selected articles were grouped in caloric restriction and fasting, enriched polyunsaturated fatty acid diets, low-fat plant-based diets, high-protein diet, elimination diets (including gluten-free, and lactose-free diets), antioxidant vitamins and minerals, fruits and fibers, prebiotics and probiotics. The dietary patterns or interventions are presented at large and commented on in the following section of the present review.

### 3.1. Caloric Restriction and Fasting

Therapeutic fasting techniques include intermittent fasting (e.g., 60% energy restriction every other day), partial fasting (e.g., 5-day diet providing 750–1100 kcal), and time-restricted feeding (limiting the daily period of food intake to ≤8 h). Therapeutic fasting leads to weight loss, it improves multiple health indices (e.g., insulin resistance and risk factors for cardiovascular disease), and reduces the proinflammatory state and oxidative stress, consecutively increasing cellular metabolism and promoting stem cell-based regeneration [81]. Intermittent fasting (IF) contributes to prevention and deceleration of chronic inflammatory diseases associated with chronic pain by a decrease in pain experience through reduction in central and peripheral inflammation [82]. Clinically, IF positively influences age-related processes [83]. A recent review also revealed the benefits of IF in improving weight control, glycemia, blood pressure and dyslipidemia; reducing oxidative stress and cardiovascular risk; improving circadian rhythms and immune responses and optimizing ketogenesis [84].

Evidence suggests that therapeutic fasting decreases peripheral and central neuronal sensitization and offers multiple health benefits for individuals living with chronic pain, [85] including extended life span, delayed age-related brain function deficits, and preserved cognitive function [86]. Neuroplastic changes in brain structure and function are a consequence of chronic pain and maintain the pain symptoms. Recent evidence showed that alterations in the brain associated with chronic pain are modifiable and reversible with effective clinical interventions (caloric restriction, intermittent fasting, exercise, incremental training). IF improves cellular functioning, increases cellular metabolism, and reduces inflammation and oxidative stress [87]. Neurobiologically, IF stimulates production of new neurons from neuronal stem cells and increases synaptic plasticity [88]. IF induces neuroendocrine activation and increases the production of neurotrophic factors [82].

Riecke et al. compared two low-energy diets to relieve symptoms of knee osteoarthritis in obese patients and also obtain weight loss by an intensive program lasting 16 weeks [46]. 192 participants were randomized to a low-energy diet or a very-low-energy diet (810 kcal/day vs. 415 kcal/day), initially comprising formula foods for the first 8 weeks, followed by a hypoenergetic diet of 1200 kcal/day for another 8 weeks. A significant reduction in pain was obtained, as evidenced by the visual analogue scale, with a weight loss of 12.8 kg (95%CI: 11.84–13.66; *p* < 0.001) in the combined groups. There were no clinically significant differences between the two diets. This 16-week program is considered a fast and efficient way to lose weight and improve symptoms in obese patients with knee osteoarthritis [46].

There is new evidence that low-carb diets restricted in FODMAPs (fermentable oligo-, di- or monosaccharides and polyols) have effects in relieving symptoms of fibromyalgia, as the longitudinal study of Marum et al. suggests [47]. Thirty-eight female participants diagnosed with fibromyalgia for approximately 10 years, entered a four-week trial with individually recommended low FODMAP dietary plans, and were instructed to fill out five different questionnaires, two related to fibromyalgia, two to assessment of pain, and one to quality of life. FODMAP intakes differed between patients, sometimes according to adherence, from as much as 25 g/day to 2.5 g/day. There were three assessment moments in the study and comparisons showed significant decreases in scores in all measured fields not only for pain associated with fibromyalgia (*p* < 0.01) but also with gastrointestinal comorbid conditions such as irritable bowel syndrome [47].

Di Lorenzo et al. reported migraine remission following ketosis in a study on 96 patients with migraine. The study demonstrated a significant improvement in headache related features including the number and frequency of headache episodes, and lower doses of specific medication. A continuous improvement for two months after stopping the diet was observed [43]. Messier et al., in a study conducted on 399 adults with knee ostheoathritis observed that after 18 months of diet and exercise, only diet groups had greater improvement in pain. The study design included a calorie-restricted diet to 800–1000 calories/day, 2 meal replacement supplements, and exercise regimen vs. intensive diet-induced weight loss vs. exercise [44].

Michalsen et al. used the Buchinger method of fasting in 48 volunteers with fibromyalgia, allotted to a usual rheumatology group or to an integrative medicine group, where alternative methods were done besides fasting. Subtotal caloric restriction of an energy intake <500 kcal/day was followed for 7 days, being preceded by 1–2 prefasting days of a 800 kcal intake consisting of monodiet of either potato, rice, or fruit. Advised fluid intake was 2–3 l, including a vegetable broth at lunch. Energy intake during the recommended 2-week fast went as low as 350 kcal/day. Foods were reintroduced stepwise. The Fibromyalgia Impact Questionnaire (FIQ) and the visual analog scale were used to evaluate pain at baseline and after 2 weeks. Both scales showed significant decrease of pain parameters, more significant in the integrative medicine group [45].

### 3.2. Enriched Polyunsaturated Fatty Acid Diets

A high-fat, low-carbohydrate, ketogenic diet can alleviate symptoms including generalized pain and headache in a wide variety of neurological diseases, e.g., brain cancer, multiple sclerosis, neurotrauma, Alzheimer, and Parkinson disease [89]. Over the past decade, investigators have demonstrated the neuroprotective role of ketogenic diet [89,90,91]. The mechanisms that contribute to the neuroprotective role of ketogenic diet include an improvement of mitochondrial function, inhibition of inflammatory mediators (interleukins and tumor necrosis factor alpha), and decrease of oxidative stress [24]. Linoleic acid is an essential polyunsaturated fatty acid that provides an endogenous substrate for linoleic acid-derived mediators of sensory signaling in the skin. Arachidonic acid and eicosapentaenoic acid play an important role as endogenous pain-relieving molecules in the central nervous system. Arachidonic acid is metabolized to both proinflammatory and anti-inflammatory eicosanoids during and after the inflammatory response [92]. Since omega-3 and omega-6 fatty acids seem to have antinociceptive and pronociceptive qualities, a 4 month randomized, single-blind, parallel-group clinical trial performed by Ramsden et al. evaluated the benefits of dietary high omega-3 and low omega-6 (H3-L6) vs. low omega-6 fatty acids supplementation in 67 ambulatory patients with chronic daily headache. Primary biochemical outcome included the HUFA score in erythrocytes (omega-6 highly unsaturated fatty acids) and analysis of omega-3 derived antinociceptive and omega-6 derived pronociceptive mediators. The H3-L6 intervention demonstrated significant improvement of the Headache Impact Test (HIT-6), the patients experiencing reduced headache pain and improved quality-of-life, accompanied with increase of antinociceptive omega-3 mediators, compared to the L-6 intervention. The L-6 intervention determined significantly decreased HUFA scores [48].

Soares et al. tested in a double-blind, randomized placebo-controlled clinical trial the benefits of omega-3 polyunsaturated fatty acids for the prevention of migraine in headache patients using amitriptyline. After 60 days, the authors noticed a reduction of days of headache in their patients and underlined the good capacity of omega-3 in prevention of migraine attacks [49].

### 3.3. Low-Fat Plant-Based Diet

A low-fat, plant-based diet has multiple benefits for health, improving glycemic control, blood pressure, blood lipid concentrations, and decreasing inflammation. Changing the lifestyle and diet of migraine patients still represents a challenge for prevention of headache crises. In a randomized, crossover intervention trial the effects of low vs. moderate lipid intake were evaluated on incidence and severity of headache. The two diets were randomly prescribed for 3 months and afterwards crossed over for the next 3 months in 83 adults with episodic or chronic migraine diagnosed according to the criteria of the International Classification of Headache Disorders. Adherence to a given diet was assessed by a food frequency questionnaire. The number and severity of attacks reported by self-assessed calendars significantly decreased in both intervention groups, being in favor of the low-lipid diet (< 20% vs. 25–30% of total daily energy intake); (*p* < 0.001 and *p* < 0.01 respectively) [51].

Another study that evaluated a plant-based diet in the management of functional limitations in chronic musculoskeletal pain was recently published by Towery et al. In an 8-week pilot study, 14 participants were instructed by a dietician to follow a plant-based diet, using a phone app to report daily food intake. The results of two self-reported questionnaires: The Numeric Pain Rating Scale and the Short Form Health Survey (SF-36) were registered at inclusion and completion of study. The intake of the plant-based diet determined significant improvements of both chronic pain and function. The researchers concluded that chronic pain can be successfully treated through a good collaboration between nutritionists and physical therapists and underlined the necessity of diet interventions in the general population, a recommendation given with caution, because their study did not involve a control group [54].

In a 20-week pilot study on patients with type 2 diabetes and painful diabetic neuropathy (*n* = 33), the investigators revealed the positive effects of a low-fat, plant-based diet in reduction of painful symptoms. The patients were randomized to intervention vs. control and asked to also take a vitamin B_12_ supplement. The diet excluded animal products and fat intake was limited to 20–30 g day. Pain scores were evaluated by the McGill pain questionnaire and the Michigan Neuropathy Screening Instrument and were significantly improved in the diet group by -8.2 points (95% CI −16.1 to −0.3, *p* = 0.04) and −1.6 points respectively (95% CI −3.0 to −0.2, *p* = 0.03) [50].

Maruki et al. studied the effects of a low-fat diet (<20 g of fat/day) in patients with upper abdominal pain of unknown cause suggestive of chronic nonalcoholic pancreatic disease [53]. According to presence of symptoms of chronic pancreatitis, 45 patients were divided into 3 groups (suggestive, indeterminate and control). After 4 weeks symptom severity was compared to inclusion on a 10-cm visual analog scale. Improvement of symptoms was compared among the 3 groups and was significantly higher in the suggestive group vs. both indeterminate and control groups (*p* < 0.001) [53]. The efficacy of a combined lacto-vegetarian diet and physiotherapy on low back pain was studied by Martínez-Rodríguez, et al. in 21 women with fibromyalgia, randomized into three groups: exercises and diet, placebo, and diet vs. control. After 4 weeks, pain assessments on the visual analog scale (EVA) scale and bioimpedance body composition revealed the most significant pain reduction and increase of muscle mass by body composition in the exercises and diet group [52].

### 3.4. High-Protein Diet

It is well recognized that adequate daily protein intake is required to preserve muscle mass and strength, especially in older patients with chronic musculoskeletal pain. Muscle wasting is a well-known occurrence in chronic pain [93]. Essential amino acids play an important role in the control of muscle anabolism and regulate protein synthesis. From essential amino acids present in high protein foods three primary neurotransmitters and pain modulators are synthesized: serotonin, endorphins, and gamma-aminobutyric acid. Endorphins are a group of endogenous opioid neuropeptides which are stronger than morphine as pain relievers. Some studies have observed that the consumption of 90–100 g of protein per day prevents neurotransmitter depletion and significant muscle-wasting [94]. A significant positive association between protein intake and pain threshold was reported among patients with fibromyalgia. Batista et al. reported that an inadequate protein intake might contribute to fibromyalgia pain by tryptophan deficiency via the tryptophan-serotonin metabolic pathway [95].

Inadequate intake of neurotransmitter precursors present in the diet, particularly essential amino acids, might lead to neurotransmitter deficiency and explain the intensity of perceived pain. Shell et al. used an amino acid blend of precursors to neurotransmitters 68405-1(AAB) in a cohort of 122 patients with chronic low back as medical food. They administered the agent in a double-blind controlled study vs. low-dose ibuprofen vs. both agents. Pain severity was quantified by the Oswestry disability scale and the Roland Morris index. Both indexes significantly improved after 28 days in the AAB and combination groups compared to ibuprofen alone, while serum levels of serine, histidine, arginine, and tryptophan increased to normal values after treatment, being directly correlated with treatment response. The substantial clinical improvement in the quality of chronic back pain was also accompanied by reduction in biomarkers of inflammation [55].

### 3.5. Elimination Diets

Elimination diets, which comprise the gluten-free diet, lactose-free diet, and histamine-free diet, are represented by eating plans in which a certain food or group of foods are removed from the diet for a given period in order to alleviate symptoms believed to be caused by them. This concept is especially used in food allergies or food intolerances triggered by natural food compounds or food additives. Symptoms manifest with considerable individual variation and may cluster to syndromes like migraine, irritable bowel, gastroesophageal reflux, or even determine autoimmune diseases [96].

Food allergies in children determine abdominal pain and diarrhea, and lymphoid nodular hyperplasia of the lower gastrointestinal tract may be diagnosed during colonoscopies. In a prospective, parallel multiarm, randomized clinical trial the relationship between food allergies and lymphoid nodular hyperplasia was studied. After excluding inflammatory or immune diseases and after a positive diagnostic colonoscopy, the investigators included 72 children with lymphoid nodular hyperplasia. The children were randomized to elimination diet, mesalamine, or symptomatic treatment for 8 weeks. Patients were followed-up 24 months. Clinical improvement was observed in all patients, without a specific effect of mesalamine or of the elimination diet compared to symptomatic treatment [57].

The effects of monosodium glutamate as dietary excitotoxin inductive of symptoms of fibromyalgia have been studied by several authors, leading to conflicting results. Holton et al. placed 57 fibromyalgia patients who also had irritable bowel syndrome on a 4-week excitotoxin elimination diet, excluding monosodium glutamate and aspartame. The diet was completed by 37 subjects, who reported significant improvement of symptoms and who were afterwards randomized to a 2-week crossover double-blind challenge with monosodium glutamate or placebo for 3 days each week. In patients on monosodium glutamate, a significant aggravation of symptoms was noted, with worsening of fibromyalgia scores as revealed by the visual analogue pain scale and the Fibromyalgia Impact Questionnaire (*p* < 0.03) vs. placebo. The authors consider that monosodium glutamate may contribute to fibromyalgia symptoms and that further research is needed to confirm their findings [56]. Vellisca et al. also assessed the effect of monosodium glutamate and aspartame elimination on perceived pain in 72 women with fibromyalgia randomized to discontinuation of monosodium glutamate and aspartame or to a waiting list. Pain was evaluated on a seven-point self-perceived scale. As compared with the previous study, no significant differences were noted between groups regarding pain at baseline or after the elimination diet. The authors concluded that the elimination diet did not improve the pain symptoms in fibromyalgia [62].

The gluten-free diet is usually indicated in gastrointestinal pathologies, such as celiac disease and IgE-mediated allergies. Gluten is the name for a mix of proteins, mainly composed by wheat grains, like gliadin and glutenin. Similar compounds can be found in barley, rye, and oats [97]. The hypotheses underlying the recommendation to remove gluten from the diet in symptomatic patients are related to wheat type. Gluten and its consequences in health is still a matter of debate. The range of symptoms related to gluten varies from absent to severe, such as celiac disease, malabsorption, and death [98]. They are a consequence of structural damage to the mucosa of the small intestine and can be reversed through a gluten-free diet [99].

Although the prevalence of gluten-related allergy and autoimmune diseases is increasing in developed countries, there is a lack of a consensus in defining the term “gluten-free” in foods [100]. The quantity of ingested wheat required for triggering symptoms and the severity of reaction varies greatly. The Food and Drug Administration in the United States and the Codex Alimentarius (http://www.codexalimentarius.net) state that gluten-free foods are edible compounds with an allowable level of gluten detectable to 20 ppm, ≤20 mg/kg in total, or processed to <100 mg/kg. Food Standards Australia New Zealand considers as gluten-free foods with no detectable gluten (<5 ppm) assessed with very specific and sensitive methods [101].

There are studies which proved that gluten consumption was associated with both gastrointestinal symptoms such as recurrent abdominal pain, bloating, altered stool consistency, and nongastrointestinal symptoms like tiredness, depression, and anxiety in patients with irritable bowel syndrome (IBS). Another cause of pain in patients who have gluten-related conditions is metabolic osteopathy. More than 75% of untreated adult celiac suffer from a loss of bone mass, and sometimes, the first symptom of celiac disease is osteoarticular pain [102]. In a case-control study on 97 IBS women with associated fibromyalgia and lymphocytic enteritis, Rodrigo et al. observed that after one year on a gluten-free diet, in selected IBS patients with fibromyalgia and positive duodenal biopsies for celiac disease, there was better outcome than in controls, with improvement of fibromyalgia impact scores, tender points, and visual analog scale. They registered a 27% improvement in perceived general health according to the SF-36 quality of life questionnaire including physical and mental scores [59].

Similar results were obtained by Slim et al. in a single-blind, 18-month, randomized clinical trial comparing the effects of gluten-free versus hypocaloric diet in patients with fibromyalgia and gluten sensitivity symptoms. Seventy-five patients with fibromyalgia were included and completed several pain-related questionnaires (Brief Pain Inventory, Revised Fibromyalgia Impact, Patient Global Impression Scale of Severity and Improvement). After 24 weeks, similar results in pain improvement were registered for both dietary interventions, which were well tolerated. Although specific in design, gluten-free diet did not decrease gluten sensitivity symptoms more than the hypocaloric diet [61].

A gluten-free diet can lead to B vitamin, fiber, iron, and trace mineral deficiencies. Half of adult celiac patients on a gluten-free diet for several years show signs of poor vitamin status that can lead to vitamin deficiency, elevated total plasma homocysteine levels, and cardiovascular disease. Thus, more attention should be given to the quality of nutrients in the gluten-free diet, especially because it is a lifelong treatment. The lack of fortified gluten-free foods is a reason for deficiencies. Another downside of gluten-free diet is represented by weight gain and obesity (usually, gluten-free foods are hypercaloric). Gluten-free diet products also have lower average protein content compared to nongluten free foods [103].

The lactose-free diet is prescribed to individuals who suffer from lactose intolerance, which affects between 30 and 50 million Americans according to the National Institutes of Health. Some people can tolerate small servings of dairy products, while others cannot digest lactose at all [104]. Lactose intolerance is a common diagnosis in patients with chronic abdominal pain. There are different genetic mutations related to the condition, especially the homozygous −13910 CC variant of the MCM-6 gene. Magiera et al. conducted a study evaluating the effects of an eight-week lactose-restricted diet on 210 patients with chronic abdominal pain for at least 12 months. 29.5% of patients were genetically positive for the homozygous T-13910-C mutation [58]. Symptoms were evaluated using a standardized gastrointestinal pain and associated symptoms questionnaire (GIQLI). The GIQLI scores significantly improved in all patients after 4 and 8 weeks of lactose-restricted diet (*p* = 0.001). While these results were logically expected in the homozygous T-13910-C mutation patients, in the remaining patients the abdominal pain relief could be explained by an unspecific effect of the diet on initial psychosomatic complaints, the authors concluded [58]. Lactose is found in many foods and it is difficult to completely avoid it from the diet. Removing lactose decreases calcium intake by 73 percent, because the main dietary source is represented by dairy products. Calcium is essential for bone growth and repair throughout the lifespan, and deficit may lead to fragile bones and osteoporosis. In both children and adults with lactose intolerance, calcium supplementation is mandatory [104]. Savaiano et al. conducted a randomized, double-blind, parallel group study in 85 patients with lactose intolerance for 35 days, investigating a novel galacto-oligosaccharide (RP-G28) vs. placebo. The study results demonstrated a significant decrease of abdominal pain and symptoms, with a better lactose tolerance at the end of treatment in the intervention group (*p* = 0.03) [60].

### 3.6. Antioxidant Vitamins and Minerals

Besides dietary advice and drug treatment, adding vitamins and minerals to healthcare plans in chronic diseases to improve the general health status of a person is a common practice worldwide [105,106].

A randomized double-blind controlled trial performed by Dhingra R et al. tested the efficacy of antioxidants vs. placebo in reducing abdominal pain in 61 patients with chronic pancreatitis. The study showed that daily doses of an antioxidant containing 600 ug organic selenium, 0.54 g vitamin C, 9000 IU b-carotene, 270 IU vitamin E, and 2 g methionine had as consequence the relief of pain after 3 months of administration [65].

Another trial tested the efficacy of a supplement containing a combination of coenzyme Q10, riboflavin and magnesium in alleviating migraine symptoms. 130 individuals aged between 18 and 65 years diagnosed as having more than 3 attacks of migraine per month were randomized to 2 groups, supplement vs. placebo. After 3 months, a reduction of the number of days with migraine, a decreased intensity of symptoms and an improvement of the scales of pain, towards statistical significance, was noted. The researchers concluded that this combination is extremely efficient for prevention of migraine attacks [66].

In a double-blinded, placebo-controlled parallel trial, Gazerani et al. addressed migraine patients randomized to 24 weeks of treatment with vitamin D3 vs. placebo. Migraine patients on vitamin-D3 presented a significant decrease in migraine frequency vs. placebo (*p* < 0.001) and also reduction of migraine days. However, pressure pain thresholds and migraine severity did not change significantly [67].

The possibilities of improving back pain also comprise administration of various supplements, especially when therapeutic options are limited [107,108]. A randomized controlled clinical trial performed for 16 weeks in 49 patients showed the benefits of Vitamin D_3_ supplementation for improvement of back pain disability in overweight or obese adults [64]. Similar to this trial, the VITAL trial was a randomized trial which used vitamin D_3_ vs. placebo to prevent worsening of musculoskeletal symptoms in 160 women with breast cancer receiving adjuvant letrozole. After 4 months, the researchers concluded that there was no change in musculoskeletal symptoms after vitamin D_3_ administration [69]. Another clinical trial performed in 2011 by Rastelli et al. showed that aromatase inhibitor-induced musculoskeletal symptoms in breast cancer can be reduced using 4 months of treatment with vitamin D_2_. The authors concluded that further investigations are necessary in this direction [70].

In an open-label single arm clinical trial, Ghai et al. investigated 68 patients with chronic lower back pain, regarding pain intensity (VAS) and functional disability (Oswestry questionnaire). All patients included were vitamin D deficient and were supplemented with 60.000 IU vitamin D3 weekly for 8 weeks. Significant reduction of pain on VAS and improvement of functional ability were observed and persisted even after 3 and 6 months after study completion [68].

In the clinical trial of Anoushirvani et al., 63 breast cancer patients receiving taxol were enrolled to 640 mg omega-3 t.i.d (gr. O) vs 300 mg vitamin E b.i.d. (gr. E) vs placebo for 3 months. According to neurological examination 28.6% patients in gr O, 33.3% in gr E and 71.4% in the placebo group presented peripheral neuropathy. At the end of the study a significant difference was noted between intervention groups and the placebo group (*p* = 0.0001), while both interventions were equally benefic [63]. Vitamin E supplementation was also benefic in improving neuropathic pain and quality of life in diabetic neuropathy patients in the clinical trial of Rajanandh et al. 92 patients were enrolled in a randomized controlled study, analyzing results of usual care vs. intervention (vitamin E 300 mg b.i.d.) and assessed by the Neuropathy Pain Score and the RAND 36 questionnaire. Significant reductions in total pain score and physical health indicators were observed after 12 weeks of treatment [71].

### 3.7. Fruits and Fibers

Abdominal pain, cramping, and bloating are frequent symptoms in constipation, which can be influenced by appropriate nutrition. In a randomized study to evaluate consequences of green banana biomass intake and laxatives, 80 children with functional constipation, abdominal pain, and painful defecation were divided into five groups, according to green banana and/or laxative intake. Consumption of green banana biomass alone produced a significant improvement of abdominal pain, painful defecation, and straining, also reducing the required doses of laxatives [73].

Fruits are known to be rich in polyphenols, which have been extensively studied for their antioxidant and anti-inflammatory properties. Inflammation is a major determinant of joint disorders, degeneration, and function loss of articular cartilage, as evidenced in osteoarthritis. Du et al. analyzed the effects on pain and inflammation of freeze-dried whole blueberries, in a randomized, double-blind trial. Sixty-three adults with symptomatic knee osteoarthritis were randomized to 40 g freeze-dried blueberry powder or placebo powder for a duration of four months. The Western Ontario McMaster (WOMAC) Osteoarthritis Index questionnaire was applied for pain assessment including difficulty to perform daily activities and stiffness. The blueberry group had significantly better results in all WOMAC scores vs. placebo with no significant changes of inflammatory markers in any group. The authors recommend daily consumption of whole blueberries to improve quality of life and reduce pain in patients with knee osteoarthritis [74].

Brain et al. examined the effect of two dietary supplement fruit juices (cherry juice and apple juice) versus placebo recommended in a personalized diet administered for 6 weeks to 42 participants with chronic pain from a Pain Clinic, who were randomized to four groups. Statistically significant improvements in pain outcomes and quality-of-life categories were obtained in both intervention groups. Brain et al. noticed a significant improvement in 3 of 5 pain scores and quality of life. The importance of the study resides in the observation that dietitian-delivered intervention contributes to relief of pain and improves dietary intake and quality of life in chronic pain [72].

Besides fruits, dietary supplements rich in fibers like partially hydrolyzed guar gum are used to improve functional abdominal pain in pediatric patients. In a randomized, double-blind pilot study Romano et al. evaluated the efficacy of the fiber supplement in 60 children with functional bowel disorders, allocated to partially hydrolyzed guar gum or placebo for 4 weeks. Severity of abdominal pain and bowel habits were evaluated with specific questionnaires. In the partially hydrolyzed guar gum group, a higher efficacy in reducing clinical symptoms and normalization of bowel habits were obtained (40% vs. 13.3%, *p* = 0.025). Complementary therapy seems to play a beneficial part in symptom control [75].

### 3.8. Prebiotics and Probiotics

Prebiotics, probiotics, and synbiotics have been extensively studied in recent years. Their properties to modify intestinal flora made them indispensable in many digestive and other pathologies [109]. It is well-known that constipation can be treated with soluble fiber supplements, macrogol laxatives without sodium sulfate, and dietary therapy. The effects of probiotics on constipation and symptoms related are not investigated enough. Cassani et al. studied in 2011 the effects of 6 weeks administration of probiotic supplementation with Lactobacillus casei Shirota in 40 Parkinson’s disease patients having also abdominal pain due to constipation [76]. The results were satisfactory with reduced abdominal pain and improvement of incomplete emptying and bloating sensations and increased number of days with normal consistency stools. As conclusions, the authors underlined the benefic effects of probiotics in normalization of abdominal pain and accompanying symptoms in individuals with constipation and Parkinson’s disease [76].

In a crossover, double-blind formula-controlled trial performed by Guerra et al., the effects of probiotics were tested during 5 weeks in 59 students suffering of chronic functional constipation, abdominal, and defecation pain. After randomization in 2 groups, one group received goat yogurt with Bifidobacterium and the other group received yogurt alone. The researchers noticed significant improvement in abdominal and defecation pain with probiotic supplementation at the end of this study [77].

It has been demonstrated that the gut microbiome might have a role in regulation of cognitive function, through complex gut-brain interactions. A double-blind, placebo-controlled, randomized controlled trial explored the cognitive and emotional Effects of Lactobacillus acidophilus vs. Lactobacillus Rhamnosus GG vs. placebo in 60 patients with fibromyalgia [78]. At the end of the study, no improvements of pain symptoms were noticed, but the researchers concluded that more studies should be developed regarding the effects of probiotics on cognitive function in the healthy population.

In 2018, the effects of NKCP^®^, a natto-derived dietary food supplement with bacillopeptidase F were tested in a double-blind, placebo-controlled, randomized crossover study performed for 4 weeks in 30 patients with neck and shoulder stiffness and headaches. The researchers showed an alleviation of headaches and chronic neck and shoulder stiffness and pain in these patients [79]. It is important to notice that no side effects were recorded. In addition, the benefic effect of NKCP^®^ on the visual analogue scale score was mentioned.

Waitzberg et al. evaluated in a randomized, double-blind, placebo-controlled study the effects of a synbiotic, combining fructo-oligosaccharides with Lactobacillus and Bifidobacterium strains (LACTOFOS^®^), given for 30 days, in 100 constipated adult women with abdominal pain. No improvement of pain symptoms was noticed at the end of this study [80]. The authors concluded that the symptoms of constipation were improved after synbiotic administration, but no change of abdominal pain was detected.

## 4. Discussion

In the last decade, numerous observational clinical studies, pilots, and RCTs have focused on dietary patterns, constituents, or supplements that influence chronic pain, trying to address pathophysiological pathways as inflammation, oxidative stress, or pharmacologic effects of nutraceuticals on pain relief. In this review we made a systematization of all diets found by our search of literature, grouping them according to caloric intake, content of basic macro- and micronutrients, elimination of allergenic foods or excitotoxins, and different food supplements. To our surprise, out of 272 results from the combined terms and filters of our search, only 59 were represented by original research, the rest of 213 being reviews, out of which 7 were meta-analyses.

In the 37 clinical trials that have corresponded to our search, the quality and intensity of pain was generally evaluated by the visual analog scale, which is a self-reported pain rating scale based on measurements recorded on a 10-cm line drawn between 0 cm no pain” and “worst pain” at 10 cm. It can be used for pain progression in a given patient or for comparison of pain between patients with identical pain pathologies [110,111,112]. The Oswestry scale was used in some studies to quantify disability in low back pain, and also the Fibromyalgia Impact Questionnaire or Numeric Pain Rating Scale for the respective condition, the Headache Impact Test (HIT-6) in migraine, and the McGill pain questionnaire or the Michigan Neuropathy Screening Instrument in neuropathic pain.

Of all pain categories defined most recently by the European Pain Federation [1], most trials referred to chronic musculoskeletal pain, i.e., low back pain (4 trials) [52,55,64,68], knee osteoarthritis in obese patients (3 trials) [44,46,74], fibromyalgia (7 trials) [45,47,56,59,61,62,78], neck pain and stiffness (1 trial) [79], musculoskeletal pain or peripheral neuropathy due to cancer treatment (3 trials) [64,69,70], chronic headache or migraine (6 trials) [43,48,49,51,66,67], generalized chronic musculoskeletal pain (2 trials) [54,72], diabetic neuropathy [50,71], taxol-induced neuropathic pain [63], or abdominal pain (10 trials) [53,57,58,60,65,71,73,75,76,77,80]. A synthesis is provided in Table 2 (Table 2).

In fibromyalgia and generalized chronic musculoskeletal pain, pain and functional deficit are alleviated by a plant-based low fat diet, a vegan diet, or a low FODMAPs diet, also positively affecting inflammatory biomarkers [47,52,54]. Eliminating MSD and aspartame from the diet, contributes to improvement of pain [56,62], while probiotics do not seem to influence chronic pain in fibromyalgia [78]. In a systematic review by Silva et al., the authors consider that existing studies to this date are low quality and not robust enough to allow conclusions about nutrition in fibromyalgia, although recent guidelines recommend a multidisciplinary approach [113]. Optimal combinations for increased efficacy to alleviate pain are pharmacological and nonpharmacological, diet being mentioned as an important tool for pain improvement [113,114]. The irritable bowel syndrome associated to fibromyalgia can be addressed by a gluten-free diet [59,61], as observed by Aman et al. [115], while neck and shoulder stiffness responds to probiotics [79].

It is considered that fibromyalgia is determined by a complex imbalance of nutrients that leads to muscle pain, i.e., mineral deficiencies of magnesium and selenium, vitamins B and D deficits, or by toxic heavy metals, i.e., cadmium, mercury, and lead. In these situations, pain inhibitory mechanisms are affected, with development of fatigue and pain symptoms, exacerbated by further decrease of bioavailability of essential nutrients. In the correct management of fibromyalgia patients, dietary guidance is considered crucial for proper correction of all vitamin and mineral deficits. Research has demonstrated that if micronutrient levels are back to normal, pain is relieved to a great extent [116]. In the review of Karras et al., present-day information derived from results of observational and supplementation studies on Vitamin D and its neuromuscular and anti-inflammatory actions, is analyzed. Recent findings have proven that hypovitaminosis D is highly prevalent in fibromyalgia. Supplementation seems to have beneficial effects on pain, but there are no specific recommendations [117].

In low back pain and knee osteoarthritis diagnosed in obese subjects, the hypocaloric diet seems to lead to adequate weight reduction, which could delay the course of disease. The effects of this diet also include lowering of cholesterol profiles and together with adjuvant interventions on cartilage metabolism they could insure a long-term alternative in the management of these patients [118]. Obesity represents an important risk factor that aggravates symptoms in knee osteoarthritis. Rios et al. identified seven systematic reviews that all included randomized trials, concluding that hypocaloric diet may increase functionality and improve quality of life in knee osteoarthritis [119].

Recently, Ebell et al. published a review of the top 20 most used therapies by primary care physicians in frequent pathologies in Canada and found that musculoskeletal pain was treated mainly with nonsteroidal anti-inflammatory drugs, diclofenac being the most effective for hip or knee osteoarthritis, while opioids in low back pain have limited use [120]. Although diet is an easily accessible option for health improvement, it is not clear whether in populations susceptible for knee osteoarthritis these effects could be obtained by diet alone, or if combination with several other methods like surgery, exercise, and drugs would bring a greater benefit [119].

A most common condition, headache alone, especially as migraine attack or as accompanying symptom in various clinical situations, seems to be responsive to low fat diets and omega3 supplements [48,49,51,121]. Short-lasting ketogenesis considerably improved the quality of headache and the number of migraine attacks in a model of one-month very-low-calorie ketogenic diet followed by a five-month standard low-calorie diet. The improvement persisted even after six-months follow-up [43]. Ketogenic diet is thought to restore brain excitability and to counteract neuroinflammation in migraine, being used with comparable efficient results in refractory pediatric epilepsy [122]. Barbanti et al. recently published a review on ketogenic diets in 150 migraneurs from case reports and prospective studies and conclude that randomized controlled studies are needed to confirm feasibility of ketogenic diets for effective prevention of episodic and chronic migraine [123]. In a very recent review published in June 2020, Gazerani citing Jahromi [124] considers that low-fat, modified Atkins, ketogenic, high-folate, and high omega-3/low omega-6 diets demonstrate beneficial effects in migraine. In addition, in this review extensive consideration is given to epigenetic diets in which environmental factors like dietary components can influence the epigenetic profile. Methylation of several genes, i.e., *COMT* (catechol-O-methyltransferase), *SLC2A9*, *SLC38A4*, and *SLC6A5* (solute carrier family 2, 38A, and 6A members), *KIF26A* (kinesin family member 26A), and *CGRP* might be associated with migraine, while specific diets addressing these alterations might work the opposite way and prevent migraine attacks [125].

Numerous clinical trials have demonstrated beneficial effects of omega3 supplementation in inflammatory and autoimmune diseases, i.e., lupus erythematosus, rheumatoid arthritis, ulcerative colitis, and also in neuroinflammatory and neurodegenerative diseases [126,127,128,129]. Neurogenic inflammation and genetic factors have an important role in the pathogenesis of migraine. In migraine headache, omega-3 fatty acids affect the inflammatory process through different mechanisms, some of which are related to modulation of eicosanoids [130]. Fischer et al. have demonstrated for the first time in man that dietary omega-3 fatty acids modulate the eicosanoid profile primarily via the CYP-epoxygenase pathway [131]. Omega-3 fatty acids, i.e., eicosapentaenoic acid (EPA) and docosahexaenoic acid (DHA) seem a feasible treatment option to alleviate migraine headache due to their potent anti-inflammatory activity and safety. A recent meta-analysis reviewing clinical studies on omega-3 in migraine, found that a high omega-3 diet or omega-3 supplements determine significant reductions in the duration of migraine attacks [132]. The most interesting comment of the authors refers to a high omega-3 diet with concomitant reduction of omega-6 fatty acids, which is also included in our review [48]. Ramsden et al. analyzed plasma samples from a randomized clinical trial testing a linoleic-acid lowering diet in severe chronic daily headache for 12 weeks. Diet-induced decrease of endogenous mediators 11 H-12 and 13E-LA was correlated with decrease in headache hours [133].

This particularly effective combination, documented by significant increases in antinociceptive omega-3 pathway markers 18-hydroxy-eicosapentaenoic acid and 17-hydroxy-docosahexaenoic acid is based on the dual benefit of increasing anti-inflammatory omega-3 fatty acids, while decreasing proinflammatory omega-6 fatty acids [113]. However, the authors consider that methodological quality is an important problem in the design of most studies included in their meta-analysis and that further randomized controlled trials are needed to include omega-3 in alternative or adjuvant treatment options in migraine [132].

Our search did not reveal dietary patterns to follow in cancer-related pain; however, two clinical trials referred to side-effects of cancer treatment, i.e., aromatase inhibitor-induced musculoskeletal symptoms in breast cancer, which are responsive to vitamin D_2_ or D_3_ supplementation, respectively [69,70], and another one to neuropathic pain secondary to taxel treatment in breast cancer, with good response to vitamin E or omega 3 supplementation [63]. Vitamin E supplementation 300 mg b.i.d. has been used with controversial results to prevent neurotoxicity induced by chemotherapy with taxanes, cisplatin, and oxaliplatin [134,135,136]. Eum et al., in a recent meta-analysis of 5 RCTs involving 319 patients, reported preventive effects of 300–600 mg/day vitamin E supplementation in chemotherapy-induced peripheral neuropathy [137]. Numerous other nutraceuticals i.e., acetyl-L-carnitine, glutathione, vitamin B6, alpha lipoic acid, and n-acetyl cysteine have been used as adjuvants in cancer treatments with neurotoxic side-effects, as analyzed in systematic literature reviews [138,139,140,141].

Diabetic complications represent a major cause of neuropathic pain and our search found two clinical trials in which either a low-fat, plant-based diet with vitamin B12 supplementation, or vitamin E supplementation alone, provided benefic results [51,73]. Diabetic sensorimotor polyneuropathy was also reported to be improved by a two month nutraceutical supplementation with 80 mg nano-curcumin vs. placebo in a parallel, double-blind randomized, placebo-controlled clinical trial conducted by Asadi et al. in 80 diabetic patients [142].

Dietary interventions in chronic visceral pain resulted from our search were related to chronic pancreatitis [53,65], in which a combination of antioxidants or a low-fat diet were recommended to relieve symptoms and to chronic constipation with abdominal pain relieved by probiotics [76,77]. In lactose intolerance abdominal pain symptoms were highly improved by a lactose-free diet or by prebiotics [58,60] Chronic abdominal pain can be caused by a variety of gastrointestinal diseases, frequently manifested through the irritable bowel syndrome. The irritable bowel syndrome is characterized by abdominal pain, bloating, and modified bowel movements aggravated by foods, most commonly fermentable oligosaccharides, disaccharides, monosaccharides, and polyols (FODMAPs) which interact with the gastrointestinal endocrine cells reducing cell densities [143]. FODMAPS and insoluble fiber increase luminal osmotic pressure in the large intestine determining bacterial fermentation that alters the microbiome, generating abdominal pain, and distension. Besides these effects of toxic dietary factors, the inherited predisposition for irritable bowel syndrome may manifest due to overlap with bacterial, viral, and parasitic infections. Consequent activation of local immune cells, dysbiosis, mucosal inflammation, and increased permeability of the intestinal wall might induce a so-called “on-celiac gluten sensitivity” with abdominal symptoms similar to gluten intolerance [144]. Therapeutic options to alleviate abdominal pain and diarrhea most frequently imply modulation of the opioid receptors present in the enteric nervous system, gelling fibers, or probiotics [145]. Specific diets such as the low-FODMAP can also relieve symptoms [146]. As shown in this review, low FODMAP intake [47] or high soluble fiber intake [75] helps reduce symptoms and improves quality of life in the irritable bowel syndrome, a finding also commented on in recent reviews [143]. Prebiotics, probiotics, and synbiotics target the gut microbiota, restoring the commensal gut microbiome, negatively affected by the lipopolysaccharides resulted from the imbalance due to pathogenic bacteria overgrowth [147]. A normally functioning gut microbiome is crucial for many aspects of human health because replacement with pathogenic microbiota not only upregulates inflammation but is also responsible for a dysfunctional gut-brain axis, promoting neuroinflammation and cognitive dysfunction and increasing vulnerability to Alzheimer’s disease [144].

Another important symptom of the irritable bowel syndrome is constipation, especially in the elderly. In this case, dysbiosis manifests by cramping and flatulence due to increased methane production, especially by *Methanobrevibacter smithii* (*M. smithii*), a methanogen bacteria abundant in the colonic flora of constipated individuals. Intervention with a probiotic mixture composed of five Lactobacilli strains and two *Bifidobacteria* strains, once daily for 60 days attenuated abdominal symptoms in the study of Seo et al., by decreasing the *Methanobrevibacter* population [148]. Similar results were reported on reducing abdominal pain due to constipation in our search, by supplementation with *Lactobacillus casei Shirota* for six weeks [76] or by probiotic goat yogurt with *Bifidobacterium* for five weeks [77]. A synbiotic, combining fructose-oligosaccharides with *Lactobacillus* and *Bifidobacterium* strains administered by Waizberg et al. for 30 days, did not induce any relief of symptoms [80]. In a recent placebo-controlled study in patients with irritable bowel syndrome, coadministration of a multistrain probiotic formulation to patients on a low FODMAP diet increased the number of Bifidobacterium species vs. placebo and is considered a promising tool for future investigation on the effects of probiotics in restoring normal colonic microbiota [149].

Rondanelli et al. evaluated the evidences of 172 studies about optimum diets in pain management and propose a food pyramid in a recent review. The base of the pyramid includes foods that should be eaten every day, namely three portions of carbohydrates with low glycemic index, five portions of fruits and vegetables, 125 mL yogurt and red wine, and also extra virgin olive oil. On the next level weekly foods are placed: four portions of legumes and fish, two portions of white meat, eggs and fresh cheese, leaving red or processed meats on top, with once per week consumption. Sweets are only allowed occasionally. The pennant at the top of the pyramid refers to specific customized supplementation for individuals with chronic pain, that includes vitamins, fiber, and omega-3 fatty acids. The food pyramid is intended to guide dietary advice for alleviating situations of chronic pain and could be used to individualize dietary interventions [150] as well as for a healthy lifestyle [72].

Limitations of study: We have limited our search to PubMed because the new upgraded version has offered the best filters for selection, as stated in the search strategy, allowing an accurate and rapid search according to our aim. It also is the most comprehensive medical database. Filters on Web of Science include “article”, “review”, “meeting abstract”, “proceedings paper”, and “editorial material”, which require further time-consuming selections and filtering. Custom year range allows only searches until 2018. In Scopus the combined search “chronic pain” and “diet” were split to “chronic”, “pain”, and “diet”, resulting in faulty redundant selections. We selected the database that seemed best to fit our purpose. The decision to limit our searches to the last decade was also made because the “Advanced search” section in PubMed offers a graph of numbers of publications in the selected field per year, and more than 80% of the articles fulfilling the criteria to be analyzed in our review were published in the last 10 years, the rest being scattered throughout the previous four decades.

## 5. Conclusions

Inflammation and oxidative stress are the main pathophysiological pathways of chronic pain [17] and most of the studies that we have found in our search have focused on diets and dietary constituents that modulate these two pathways in order to alleviate pain.

Individual responsibility to keep up with the diet and individual dietary guidance by specialized nutritionists is recommended in order to obtain long-lasting results. Dietary patterns, targeted diets, and interventions found in the selected articles were grouped according to diet type or supplementation type. Most of the studies, even if small in size, have revealed positive results. We have compared them with the negative ones in the same field of intervention. In most of the studies, limitations are represented by the small numbers of subjects included, flawed design, or lack of repeatability and feasibility of diets for longer periods of time. On the other hand, the dietary interventions offer an optimistic perspective for future research as we progress in understanding the complex interactions in human nutrition. Future perspectives of diet therapy in chronic pain are related to intracellular signaling pathways and molecular mechanisms that influence pain development. In the near future, miRNAs could be used as biomarkers of pain and new research could tackle the role of miRNAs in influencing chronic inflammation and alleviating pain [151]. Besides functional disability, chronic pain is also associated with significant emotional distress. It is necessary to perform a complex biopsychosocial evaluation to better understand chronic pain, leading to a multifactorial diagnosis obtained through interconnected biological, psychological, and social factors, which inevitably influence the adherence to any nutritional intervention and also its outcomes [8]. In conclusion, dietary approaches to chronic pain should include lifestyle education for a balanced diet following healthy eating patterns, interdisciplinary care, and food supplements where needed to obtain the best results.

## Figures and Tables

**Figure 1 nutrients-12-02510-f001:**
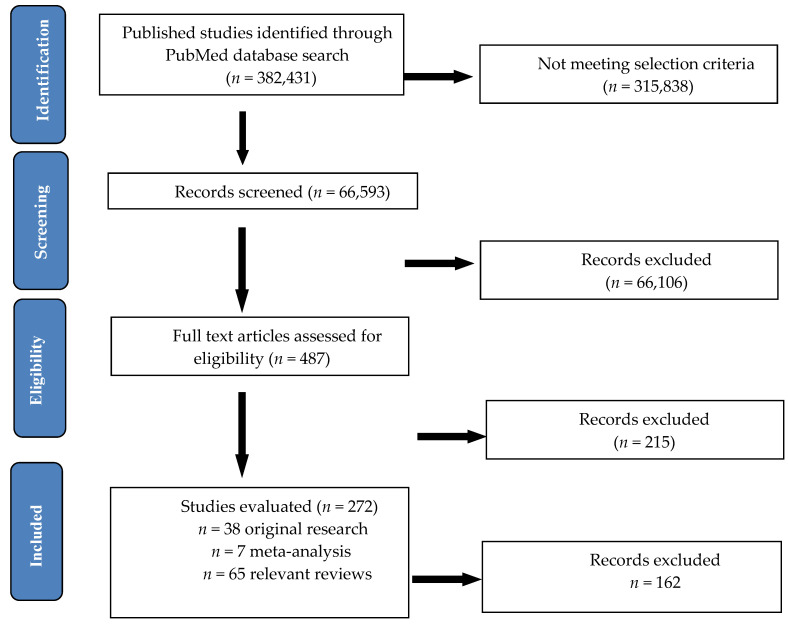
Flow diagram of the literature search process.

**Table 1 nutrients-12-02510-t001:** Clinical trials and RCTs (Randomized Control Trials) evaluating the effects of various diets on chronic pain syndromes.

Diet Type	Number	Authors and Year	Type of Study	Duration of Study	Number of Patients	Chronic Pain Condition	Diet Intervention	Effect on Pain
						Primary/Secondary Outcomes		
Caloric restriction and fasting	1.	Di Lorenzo et al. Eur. J. Neurol. 2015 [43]	two parallel groups, proof-of-concept study	12 months	96 subjects (96 female)	Headache due to starvation or a ketogenic diet	1-month very-low-calorie ketogenic diet prescription followed by a 5-month standard low-calorie diet vs. 6-month standard low-calorie diet	Migraine improvement during short lasting ketogenesis from 1 month
						Primary outcomes		
	2.	Messier et al., JAMA, 2013 [44]	single-blind randomized	18 months	399 subjects (325 female and 74 male)	Knee osteoarthritis in overweight and obesity	Calorie-restricted 800–1000 calories/day, 2 meal replacements supplements and exercise regimen vs. intensive diet-induced weight loss vs exercise	reductions in knee compressive force, weight loss and less pain in diet group and exercise regimen
						Primary outcomes		
	3.	Michalsen A et al., Evid Based Complement Alternat Med., 2013 [45]	controlled, nonrandomized pilot study	12 weeks	48 subjects (46 female and 2 male)	General body pain due to fibromyalgia	Conventional Medicine vs. Integrative Medicine Including Fasting Therapy	Improvements of pain scores after 2 weeks
						Primary outcomes		
	4.	Riecke et al., Osteoarthritis Cartilage, 2010 [46]	prospective, pragmatic randomized clinical trial, with blinded outcome assessors	16 weeks	192 subjects (155 female and 37 male)	knee osteoarthritis in obese patients	8 weeks of low-energy diet (LED; 810 kcal/day) or a very-low-energy diet (VLED; 415 kcal/day) using formula foods	weight loss and highly significant improvements in pain symptoms
						Primary outcomes		
	5.	Marum et al., Scandinavian Journal of Pain, 2016 [47]	pilot, open label, randomized clinical trial	4 weeks	38 subjects 38 female)	General body pain in female patients diagnosed with fibromyalgia	low fermentable oligo-di-monosaccharides and polyols (FODMAP) diet	significant reduction in gastrointestinal disorders and fibromyalgia symptoms, including pain scores
						Primary outcomes		
Enriched polyunsaturated fatty acid diets	1.	Ramsden CE et al. Pain. 2013 [48]	randomized, single-blind, parallel-group clinical trial	12 weeks	67 subjects (58 female and 9 male)	chronic daily headaches	dietary high n-3 and low n-6 fatty acids supplementation	reduced headache pain, improved quality-of-life
						Primary outcomes		
	2.	Soares AA et al., Nutr Neurosci. 2018 [49]	prospective, experimental, controlled, double-blind	60 days	51 subjects (36 female and 15 male)	Chronic migraine	Omega-3 dietary supplements vs placebo	reduction of days of headache and prevention of migraine attacks
						Primary outcomes		
Low-fat plant-based diet	1.	Bunner AE et al., Nutr Diabetes. 2015 [50]	randomized parallel aassignment	20 weeks	33 subjects (19 female and 14 male)	painful diabetic neuropathy in type 2 diabetes patients	a low-fat, plant-based diet in combination with a vitamin B12 supplement vs control group (only with vitamin B12 supplement)	improvement of clinical and pain symptoms, pain scales and quality of life
						Primary outcomes		
	2.	Ferrara LA et al., Nutr Metab Cardiovasc Dis. 2015 [51]	randomized, crossover intervention trial	6 months	83 subjects (63 female and 20 male)	migraine crises	two dietary regimens a low-lipid vs. normal-lipid diet	Reduced numbers of crises and severity of pain, with a significant difference in favor of the low-lipid diet.
						Primary outcomes		
	3.	Martínez-Rodríguez et al., Nutr Hosp. 2018 [52]	randomized, placebo-controlled study	4 weeks	21 subjects (21 female)	lower back pain in women with fibromyalgia	lacto-vegetarian diet and stabilization core exercises vs placebo + lacto-vegetarian diet vs control	pain reduction and improved body composition in first group
						Primary outcomes		
	4.	Maruki J et al., Pancreas. 2013 [53]	randomized, controlled trial	4 weeks	45 subjects	upper abdominal pain in nonalcoholic mild pancreatic disease	low-fat diet (<20 g of fat/day)	improvement of visual analog scale score
						Secondary outcomes		
	5.	Towery et al., Complement Ther Med, 2018 [54]	randomized, controlled trial	8 weeks	14 subjects	Chronic musculoskeletal pain	plant-based diet consisted of grains, fruits, vegetables and legumes	decreased pain and improvement and quality of life
						Primary outcomes		
High-protein diet	1.	Shell, Am J Ther., 2016 [55]	double-blind controlled study	28 days	122 subjects	chronic lower back pain	ibuprofen alone (400 mg daily) vs amino acid blend alone (two 355 mg capsules twice daily) vs the combined use of ibuprofen (400 mg daily) and amino acid blend (two 355 mg capsules twice daily)	substantial improvement in chronic back pain
						Primary outcomes		
Elimination diets	1.	Holton et al., Clinical experimental rheumatology, 2012 [56]	double-blind, placebo-controlled, cross-over clinical trial	4 weeks	37 subjects (34 women and 3 male)	General body pain in patients with fibromyalgia with irritable bowel syndrome	diet excluding monosodium glutamate vs. placebo	improvement of pain symptoms and tender point number
						Secondary outcomes		
	2.	Lucarelli S et al., Pediatr Allergy Immunol. 2015 [57]	prospective, parallel multiarm, randomized clinical trial	8 weeks	72 children	Abdominal pain due to intestinal lymphoid nodular hyperplasia in children	Elimination diet vs. mesalamine vs. symptomatic treatment	diet had no effect on pain symptoms compared to symptomatic therapy
						Secondary outcomes		
	3.	Magiera R et al., Clin Lab. 2014 [58]	Nonrandomized, Two Armed Intervention Study without Control Group	2 months	210 subjects	Chronic abdominal pain in lactose intolerance	Lactose restricted diet	improvement of abdominal pain symptoms
						Primary outcomes		
	4.	Rodrigo L et al., Arthritis Res Ther. 2014 [59]	case-control study	1 year	229 subjects (197 female and 32 male)	General body pain in irritable bowel syndrome plus fibromyalgia with/without lymphocytic enteritis	gluten-free diet	significant improvement in all symptoms and pain scales in group with lymphocytic enteritis
						Primary outcomes		
	5.	Savaiano DA AT et al., Nutr J. 2013 [60]	randomized, double-blind, parallel group, placebo-controlled study	66 days	61 subjects	Abdominal pain in lactose intolerance	RP-G28 novel galacto-oligosaccharide (GOS) vs placebo (corn syrup)	reduction in abdominal pain and improve of all symptoms of the lactose intolerance
						Primary outcomes		
	6.	Slim M et al., J Clin Gastroenterol 2017 [61]	pilot, open-labe, randomized clinical trial	24 weeks	75 subjects (73 female and 2 male)	General body pain in patients with fibromyalgia experiencing gluten sensitivity symptoms	gluten-free diet vs. hypocaloric diet	similar beneficial outcomes in alleviating pain
						Secondary outcomes		
	7.	Vellisca MY et al., Rheumatol Int. 2014 [62]	case-control study	3 months	72 subjects (72 female)	General body pain due to fibromyalgia	discontinuation of dietary monosodium glutamate and aspartame vs waiting list	no improvement of pain symptoms
						Primary outcomes		
Antioxidant vitamins and minerals	1.	Anoushirvani AA. et al., Open Access Maced J Med Sci. 2018, [63]	randomized, placebo-controlled study	3 months	63 Subjects (46 female and 17 male)	Paclitaxel-induced peripheral neuropathy	640 mg omega-3 three times a day Vs 640 mg omega-3 three times a day Vs placebo	vitamin E and omega-3 may greatly enhance quality of life
						Secondary outcomes		
	2.	Brady SRE et al., Steroid Biochem. Mol. Biol. 2019 [64]	randomized, placebo-controlled study	16 weeks	49 subjects (18 female and 31 male)	low back pain in overweight or obese adults with vitamin D deficit	bolus oral dose of 100,000 IU followed by 4000 IU cholecalciferol/day vs. placebo	improvement of back pain disability in subjects with vitamin D deficit
						Primary outcomes		
	3.	Dhingra R et al., Pancreas. 2013 [65]	randomized, placebo -controlled trial	3 months	61 subjects (18 female and 43 male)	Abdominal pain in chronic pancreatitis	Antioxidants vs. placebo (Antioxidants supplements daily doses of 600 ug organic selenium,0.54 g vit C, 9000 IU b-carotene, 270 IU vit E and 2 g methionine)	pain relief
						Primary outcomes		
	4.	Gaul et al., J Headache Pain. 2015 [66]	randomized, placebo-controlled, parallel-arm, double-blind, prospective multicenter study	12 weeks	112 subjects (97 female and 15 male)	migraine crises in adults under 65 years	Multivitamins: 400 mg B2, 600 mg Mg,150 mg Q10, 750 ug vitamin A, 200 mg vitamin C, 134 mg vitamin E, 5 mg B1, 20 mg B 3,5 mg B 6,6 ug B12, 400 lg B 9,5 ug vitamin D, 10 mg B5, 165 ug B 7, 0.8 mg Fe, 5 mg Zn, 2 mg Mn, 0.5 mg Cu, 30 lg Cr, 60 ug Mo, 50 ug Se,5 mg bioflavonoids vs. placebo	Improvement of migraine pain and no reduction of migraine days
						Primary outcomes		
	5.	Gazerani et al., Curr Med Res Opin. 2019 [67]	randomized, double-blinded, placebo-controlled, parallel trial	28 weeks	48 subjects (36 female and 12 male)	Migraine in adults	100 μg/day D3-Vitamin vs placebo	Improvement only of migraine frequency
						Primary outcomes		
	6.	Ghai B et al., Pain Physician. 2017 [68]	open label, single arm clinical trial	6 months	68 subjects (31 female and 37 male)	chronic low back pain in adults with insufficient or vitamin D deficit	60,000 IU oral vitamin-D3 supplementation every week for 8 weeks	improvement of pain intensity and functional disability
						Primary outcomes		
	7.	Khan et al., Breast cancer research treatment, 2017 [69]	randomized, placebo-controlled trial	4 months	160 subjects (160 female)	Musculoskeletal pain due to breast cancer	30,000 IU oral VitD3/week + daily supplement of 1200 mg calcium and 600 IU vitamin D3 vs placebo + daily supplement of 1200 mg calcium and 600 IU vitamin D3	no change of musculoskeletal symptoms
						Primary outcomes		
	8.	Rastelli et al., Breast cancer research treatment, 2011 [70]	double-blind placebo-controlled randomized phase II trial	4 months	60 subjects (60 female)	Musculoskeletal symptoms in breast cancer induced by aromatase inhibitor	50,000 IU Vitamin D2 vs. placebo	improvement of musculoskeletal symptoms, pain scores and severity improved
						Primary outcomes		
	9.	Rajanandh et al., Pharmacol. Rep. 2014 [71]	controlled randomized trial	12 weeks	92 subjects	diabetic neuropathy	vitamin-E 300 bid	reduction in total pain score in all questionnaires applied
						Primary outcomes		
Fruits and fibers	1.	Brain K et al., Nutrients. 2019 [72]	controlled randomized trial	6 weeks	60 subjects (41 female and 19 male)	Generalized chronic musculoskeletal pain	personalized dietary consultations and active fruit juice vs personalized dietary consultations and placebo fruit juice vs waitlist control group and active fruit juice vs waitlist control group and placebo fruit juice	significant improvement in 3 of 5 pain scores and quality of life in dietary intervention groups
						Primary outcomes		
	2.	Cassettari VMG et al., J Pediatr (Rio J.). 2019 [73]	prospective, interventional, randomized clinical study	8 weeks	80 subjects (43 female and 37 male)	Abdominal pain due to functional constipation in children and adolescents	green banana biomass alone vs green banana biomass plus PEG 3350 with electrolytes vs green banana biomass plus sodium picosulfate vs PEG 3350 with electrolytes alone vs sodium picosulfate alone	alleviation of abdominal pain and pain defecation by adding green banana biomass
						Secondary outcomes		
	3.	Du et al., Nutrient, 2019 [74]	randomized, double-blind trial	4 months	49 subjects (35 female and 14 male)	knee osteoarthritis	Blueberry powder vs. placebo powder	pain and quality of life improvement
						Primary outcomes		
	4.	Romano C et al. World J. Gastroenterol. 2013 [75]	randomized double-blind pilot study	8 weeks	60 subjects (37 female and 23 male)	chronic abdominal pain due to irritable bowel syndrome in pediatric patients	Partially hydrolyzed guar gum vs. placebo	tendency toward normalization of bowel habit and pain control
						Primary outcomes		
Prebiotics and probiotics	1.	Cassani E et al., Minerva Gastroenterol. Dietol. 2011 [76]	pilot study	6 weeks	40 subjects	Abdominal pain due to constipation in Parkinson disease	Probiotic supplementation with Lactobacillus casei Shirota	improvement of abdominal pain, decreased bloating, normalization of stools
						Primary outcomes		
	2.	Guerra PV et al., World J. Gastroenterol. 2011 [77]	crossover, double-blind formula controlled trial	5 weeks	59 subjects (47 female and 12 male)	Chronic abdominal pain due to functional constipation and defecation in students	Probiotic goat yogurt with Bifidobacterium vs. yogurt alone	significant improvement in abdominal and defecation pain
						Primary outcomes		
	3.	Roman et al., Scientific Reports, 2018 [78]	double-blind, placebo-controlled, parallel assignment	8 weeks	60 (28 female and 3 male)	General body pain due to fibromyalgia	Lactobacillus acidophilus vs. Lactobacillus Rhamnosus GG vs. placebo	no improvement of pain symptoms
						Secondary outcomes		
	4.	Sunagawa Y et al., Biol. Pharm. Bull. 2018 [79]	double-blind placebo-controlled randomized crossover study	4 weeks	29 subjects (8 female and 21 male)	neck and shoulder stiffness, headaches	250 mg of NKCP^®^, a natto-derived dietary food supplement with bacillopeptidase F vs placebo	alleviation of headaches and chronic neck and shoulder stiffness and pain
						Primary outcomes		
	5.	Waitzberg DL et al., Clinical Nutrition, 2013 [80]	randomized, double-blind, placebo-controlled study	30 days	100 subjects (100 female)	abdominal pain due to chronic constipation	synbiotic, combining fructooligosaccharides with Lactobacillus and Bifidobacterium strains (LACTOFOS^®^) vs. maltodextrin (placebo group)	no improvement of pain symptoms
						Primary outcomes		

**Table 2 nutrients-12-02510-t002:** Synthesis of dietary patterns and interventions in chronic pain types.

Chronic Pain Category.	Chronic Pain Type	Dietary Pattern/Intervention	References
Chronic musculo-skeletal pain	Fibromyalgia	Low FODMAPs diet	[47]
		Elimination diet (MSD and aspartame)	[56,62]
		Gluten-free	[59,61]
		Fasting	[45]
		Probiotics	[77]
	Low back pain	Lacto-vegetarian diet	[52]
		High-protein diet (amino acids supplementation)	[55]
		Vitamin D3 supplementation	[64,68]
	Knee osteoarthritis in obese patients	Calorie-restricted diet	[44,46]
		Blueberry polyphenols supplementation	[74]
	Neck pain and stiffness	Probiotics	[79]
	Musculoskeletal pain due to breast cancer treatment	Vitamin D2/D3 supplementation	[69,70]
	Generalized chronic musculoskeletal pain	Plant-based low-fat diet	[54]
		Fruit juice (apple/cherry)	[72]
Chronic headache	Chronic headache or migraine	Very-low-calorie ketogenic diet	[43]
		Fatty acids supplementation (high n-3 and low n-6)	[48]
		Omega-3 supplementation	[49]
		Low-fat diet	[51]
		Multivitamins and vitamin D3 supplementation	[66,67]
Neuropathic pain	Diabetic neuropathy	Low-fat, plant-based diet with vitamin B12 supplementation	[50]
		Vitamin E supplementation	[71]
	Taxol-induced neuropathic pain	Vitamin E or omega-3 supplementation	[63]
Chronic abdominal pain	Upper abdominal pain in pancreatic disease	Low-fat diet	[53]
		Antioxidants supplementation	[65]
	Intestinal metaplasia in children	Lactose elimination diet	[57]
	Lactose intolerance	Lactose elimination diet	[58]
		Novel food RP-G28 galacto-oligosaccharide	[60]
	Functional constipation	Increased fibers	[73]
		Probiotics/synbiotics	[76,77,80]
	Irritable bowel syndrome	Increased fibers	[75]

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
