# Peer review of "Dietary Patterns and Interventions to Alleviate Chronic Pain"

_nutrients, 2020, doi:10.3390/nu12092510_

Round 1
Reviewer 1 Report
This review is on an interesting topic of diet and chronic pain. However, I have several reservations in the way this review has been put together and presented. The authors need to justify why the database search was restricted to PubMed alone. I suspected other relevant databases will yield additional primary data not listed in PubMed. In Figure 1, the authors need to also briefly specify the reasons for exclusion of various studies at the various stages of the literature selection process. The figure should stand alone from the text. The 'Introduction' of the review also principally delves into classification of chronic pain without delving into the existing background on nutritional factors in alleviating chronic pain. As authors already eluded to in the Discussion section, several review papers on this topic already exist and therefore justification needs to be made on how this review provides any novelty as compared with existing reviews. The only table of data presented appear rather scattered. The listing of the effects in particular need to be quantified in greater detail. I suggest that the tables be split into the relevant sub-sections within the manuscript. Furthermore, the authors mainly listed dietary components with favourable effects on chronic pain. For example while talking about dietary fats, only effects of n-3 fatty acids to reduce chronic pain has been discussed without referring to other proinflammatory fatty acids (e.g., arachidonic acid) which may increase chronic pain. The same goes for ketogenic diets which has been associated with increasing some chronic pains (e.g., chronic headaches). Therefore while the authors discuss that most of the studies have shown "positive effects" to intervention, I suspect that many of the studies showing the contrary have not been listed. The various studies also measured variable outcome measures using inconsistent methods. This should be discussed in greater detail as the limitation of this review. In several places the interpretation of the data appear minimalistic (e.g., Line 650) or unclear (e.g., Lines 495-496).
Author Response
Reviewer 1:
- We have limited our search to PubMed because the new up-graded version has offered the best filters for selection, as stated in the search strategy, allowing an accurate and rapid search according to our aim. It also is the most comprehensive medical database. Filters on Web of Science include “article”, “review”, “meeting abstract”, “proceedings paper” and “editorial material”, which require further time-consuming selections and filtering. Custom year range allows only searches until 2018. In Scopus the combined search “chronic pain” and “diet” were split to “chronic”, “pain” and “diet”, resulting in faulty redundant selections. This was an invited review with a very tight dead-line. We selected the database that seemed best to fit our purpose.
- The decision to limit our searches to the last decade was also made because the “Advanced search” section in PubMed offers a graph of numbers of publications in the selected field per year, and more than 80% of the articles fulfilling the criteria to be analyzed in our review were published in the last 10 years, the rest being scattered throughout the previous 40 decades and implying a sorting effort for which we did not have enough time.
- We have rewritten the introduction adding more observations on diet and nutrients integrating the types of pain in the context of diets, dietary constituents and supplements. We have provided a background of current knowledge in the field.
- We have proof-read the whole article and we have made all necessary corrections of English language
- We have realigned the figure and the table text.
- We have remade Table 1 entirely. We have re-labeled “Conditions/Comorbidities” to “Chronic Pain Condition.” We have reviewed once more the studies included to validate the chronic pain context. We have added another column to “Outcomes” to indicate if the chronic pain condition in the specific study was a primary or secondary clinical outcome. We have regrouped the studies in 8 dietary intervention categories: Caloric restriction and fasting, Enriched polyunsaturated fatty acid diets, Low-fat plant-based diets, High-protein diets, Elimination diets, Antioxidant vitamins and minerals, Fruits and fibers and Prebiotics and probiotics in the table and in the text. For each study, we have clearly specified the pain condition and comorbidities in the respective rows.
- By request of all reviewers we have eliminated the DASH trial (low sodium) because it investigated daily headache risk in otherwise healthy subjects, which is not chronic pain. Also the SU.VI.MAX trial which evaluated the effect of dietary vitamins and minerals on function-limiting pain in older adults, which is not explicitly chronic pain, was excluded from our review. We have entirely removed the Personalized Lifestyle and Healthy Lifestyle” section and have incorporated trials related to diet composition into other sections. We checked again the trials previously included in this section, and we removed the ones that made no reference to diet composition. Omega-3 fatty acid studies were placed under Enriched polyunsaturated fatty acid diets, instead of being kept under Antioxidant vitamins and minerals.
- We have commented on the aspect of anti-inflammatory vs proinflammatory supplements in discussions
We have included the study limitations and have rewritten the conclusions

Reviewer 2 Report
A comprehensive review of dietary patterns and chronic pain is needed in the field, and is a large ambitious undertaking. The authors used a search strategy that limited the available literature to just over 100 studies including original research, meta-analyses and reviews. They then analyzed the articles according to diet-type with a discussion based on the type of chronic pain. There are many issues with the approach and execution throughout that need to be addressed in order to create a valuable well-written piece of literature for the field.
Primary critiques
- The structure of the review as a whole needs to be addressed to allow for an easy-to-read article that follows a logical sequence. One suggestion would be to rewrite the introduction (see below) and then decide to either base the review on diet type or on pain type. The discussion could then be used to address the areas of overlap, the diets and evidence that support each other (i.e. anti-inflammatory or antioxidant studies) as well as the conflicting diets/evidence (i.e. how high fat and low fat diets both claim to improve pain scores). The conclusion would then work as a summary similar to its current format as well as an opportunity to discuss where the research needs to go in the future to clarify many of the contradiction and lack of clinical recommendations. This could be an opportunity to discuss the possible need for integration of individual metrics in order to provided personalized dietary advice for optimum improvement of pain and/or other health outcomes.
- The introduction in its current format is not particularly relevant to the topic of the paper and does not discuss diet until the last paragraph. This should be rewritten to discuss diet and nutrients more thoroughly. Much of the discussion of chronic pain, types of pain, prevalence and risk groups was either not relevant or not put into context with diet appropriately. There is a lot of room for improvement in the introduction.
- Many studies were not included in this review that are relevant because of the search and exclusion criteria. A 10 year cutoff for instance is not an appropriate approach as diet studies can be multi-year endeavors that do not lose their relevance years later. Many studies on ketogenic diets, paleo diets, Mediterranean diets, diets in neuropathic pain (i.e. diabetic neuropathy) and more were left out of the review.
- Multiple times review articles are cited during the description of original research. For instance, you write in lines 649-650 that “research has demonstrated…” and then cite a review article and not the original research.
- Lines 684-789: This is the best section of the entire review. It flows well and is well constructed. Although there are still multiple grammatical issues that need correcting within this section, this part is a significant improvement from the rest.
- If you are going to summate all of the dietary recommendations from the literature into a single approach (the pyramid in citation 115), then you should include a figure providing a clear visualization in the conclusion. Although, it may be better to develop unique conclusions based on your review that diverge or go beyond what is suggested in the Rondanelli article, unless you completing agree with them (in which case the value of your article may be decreased).
Minor Critiques
- Figures and table text have multiple alignment issues.
- Lines 198-203 How is this a "personalized diet"?
- 204-209: You do not say why this is in the calorie restriction section or how it is relevant with so many different interventions.
- 210-212: How is acupuncture and lifestyle advice indicative of the role of diet in chronic pain?
- 306-313: You have this study in the “low-fat diet” group although there is no breakdown of macronutrients in the article to verify the amount of fat before or after dietary intervention.
- You say in lines 624-628 that the studies you found use the visual analog score, although many use more objective measurements and the were not discussed in the review.
- 629-632: You only categorize 23 of the 46 trials. Where do the other 23 fall?
- 644-650: What is written here is a minimally altered reproduction of the abstract from citation 93.
- 660-666: This study does not necessarily advocate that a specific diet benefits osteoarthritic pain directly. It more suggests that weight loss from any means will lead to less mechanical stress on the joints.
Author Response
- We have rewritten the introduction adding more observations on diet and nutrients integrating the types of pain in the context of diets, dietary constituents and supplements. We have provided a background of current knowledge in the field.
- We have proof-read the whole article and we have made all necessary corrections of English language
- We have realigned the figure and the table text.
- We have remade Table 1 entirely. We have re-labeled “Conditions/Comorbidities” to “Chronic Pain Condition.” We have reviewed once more the studies included to validate the chronic pain context. We have added another column to “Outcomes” to indicate if the chronic pain condition in the specific study was a primary or secondary clinical outcome. We have regrouped the studies in 8 dietary intervention categories: Caloric restriction and fasting, Enriched polyunsaturated fatty acid diets, Low-fat plant-based diets, High-protein diets, Elimination diets, Antioxidant vitamins and minerals, Fruits and fibers and Prebiotics and probiotics in the table and in the text. For each study, we have clearly specified the pain condition and comorbidities in the respective rows.
- By request of all reviewers we have eliminated the DASH trial (low sodium) because it investigated daily headache risk in otherwise healthy subjects, which is not chronic pain. Also the SU.VI.MAX trial which evaluated the effect of dietary vitamins and minerals on function-limiting pain in older adults, which is not explicitly chronic pain, was excluded from our review. We have entirely removed the Personalized Lifestyle and Healthy Lifestyle” section and have incorporated trials related to diet composition into other sections. We checked again the trials previously included in this section, and we removed the ones that made no reference to diet composition. Omega-3 fatty acid studies were placed under Enriched polyunsaturated fatty acid diets, instead of being kept under Antioxidant vitamins and minerals.
We have included the study limitations and have rewritten the conclusions

Reviewer 3 Report
Journal: Nutrients (ISSN 2072-6643)
Manuscript ID: nutrients-868395
Type: Review
Number of Pages: 31
Title: Dietary patterns and interventions to alleviate chronic pain
Authors: Simona Dragan, Maria-Corina Șerban*, Georgiana Damian*, Ruxandra Christodorescu
Overall Recommendation: Reconsider after Major Revisions
Brief Summary: This is an ambitious review article focusing on the role of diet and its many facets on chronic pain conditions. It is an important topic of consideration for both the pain and nutrition fields, but also the broader medical community. The authors utilized a keyword-based filter strategy with the PubMed NCBI database to identify clinically-relevant original studies investigating the effects of various diet interventions on chronic pain outcomes. The authors sub-categorized all identified studies into specific sections ranging from caloric restriction to probiotics. It highlights several important clinical findings whereby different chronic pain conditions were reduced through controlled dietary intervention.
Broad Comments:
Strengths
- Identified and addressed an increasingly important intersection of research between diet/nutrition and chronic pain.
- Utilized PubMed database to identify relevant original clinical research articles for review.
- Despite great heterogeneity amongst the articles reviewed, the authors managed to categorize all the research articles in an accessible format while also addressing several different areas of diet and any interventional role (or lack thereof) against many different chronic pain conditions.
- The authors offered some interesting insight and important take-aways in their discussion of the literature review.
Weaknesses
- The interconnection between diet/nutrition and chronic pain experience is becoming more apparent owing to increasing preclinical and clinical research efforts. What the fields need at this point is emphasis on the constituents of specific diets. Sometimes, this might not actually be the primary dietary constituent being tested in a study. However, important details pertaining to the compositions of tested diets were frequently omitted. It is understandable if a specific study did not provide much information, but there were many studies that provided substantial information on the dietary intervention that were not considered here. For example, the authors highlight a caloric restriction study that used a low-energy diet and a very low-energy diet, both of which reduced significantly knee osteoarthritis pain. Thus, a most important question is with a calorie-restricted diet, what dietary content was included in these low-energy diets? Did any meet minimum dietary recommendations?
- Although the authors specifically stated that the review was considering interventional diets for chronic pain conditions, several studies were included that did not fit the criteria. For example, the DASH trial (low sodium) investigated daily headache risk in otherwise healthy subjects, which is not chronic pain. Another example is the SU.VI.MAX trial which considered the effect of dietary vitamins and minerals on function-limiting pain in older adults, which is not explicitly chronic pain. Including these two, there were several other examples as well (listed below), which invokes the question as to how closely some of the studies were reviewed before being included in this manuscript.
- As a lengthy review was just published in Nutrients on migraine and diet (Gazerani, 2020), perhaps inclusion of migraine studies in this review is redundant?
- Weaknesses to the filtering strategy should be discussed (or better, changed) as many pertinent clinical studies were missing, likely because they were arbitrarily filtered out. For example, Ramsden et al., Sci Signaling, 2017 or Gazerani et al., Curr Med Res Opin, 2018 are not in the reference list. Also, Nutrients just published a review on migraine and diet in May 2020 by Parisa Gazerani. There are many references to original diet-migraine trials in that review that are not referenced in this review. If they were not filtered out, is there a reason they were not included?
- Peripheral neuropathies include a huge cross-section of patients that experience chronic pain for years. However, studies entailing dietary interventions against peripheral neuropathies are almost exclusively missing in this review. Peripheral neuropathies certainly qualify as chronic pain. There have been substantive clinical research investigating, for example, diet against diabetic neuropathy. Peripheral neuropathy was also defined in the introduction. Is there a reason those studies are missing?
- There are several studies that would better fit under a different category. Omega-3 fatty acid interventional studies should go under “High-fat diet” or perhaps an entirely new category could be made, such as “Enriched polyunsaturated fatty acid diets.” While I generally approve of the current list of diet categories, perhaps they might be better organized under more specific designations? Also, I think it would truly benefit the review if the “Personalized Lifestyle and Healthy Lifestyle” section was removed and the trials included were incorporated into other sections, specifically those related to the diet composition.
- For Table 1 – I recommend re-labeling “Conditions/Comorbidities” to “Chronic Pain Condition.” I think the studies included in Table 1 need to be reviewed once more to validate the chronic pain context. Also, more specific details on conditions/comorbidities are needed to make clear of this. For example, for studies classified as headache, what type specifically (migraine, cluster headaches, etc)? Perhaps consider adding another column “Outcome” and indicate if the chronic pain condition in the specific study was a primary or secondary clinical outcome. It might benefit the reader too if studies in Table 1 were ordered according to the categories discussed below, rather than alphabetical by first author.
- There are some concerning duplications of previously published literature that need to be addressed, at the very least with citations. For example, Lines 162-163 read “Clinically, IF attenuates maladaptive age-related processes and disease.” The exact same sentence can be found in Sibille et al., 2016, J Pain who cited Longo, 2013.
Specific Comments:
Line 104 – What is the justification for focusing only on studies published in the past 10 years?
Line 150 – The first entry in Table 1 (reference 44) reported the frequency and intensity of headaches in subjects involved in a controlled diet study, however these were not chronic headache pain patients. This being a post-hoc study of the DASH trial, it specifically states that reported headaches (regardless of frequency or intensity) were a side effect of the diet. They also state that no baseline headache information was available for subjects enrolled in the DASH study. Ergo, this specific study should be excluded as it does not evaluate diet in the context of a chronic pain condition, per the aim of the study. Since the authors provided nice introductory context on the current definitions of various chronic pain conditions, I would ensure that studies included in this review correspond to those definitions.
Line 154: Consider changing to “Caloric Restriction and Fasting” – therapeutic fasting is not the same as caloric restriction. Subjects on a time-restricted therapeutic fasting-based diet may still consume the equivalent calorie content per day, whereas caloric restriction consistently and persistently reduces daily average calorie intake.
Line 162 – reference needed.
Line 163 – reference needed.
Lines 167-169 – I would consider the chief complaints of chronic pain patients being the physical and emotional toll of the chronic pain condition and reduced quality of life. The aforementioned “benefits” of extended life span and reduced cognitive deficits related to aging seem unrelated. In the brain of a chronic pain patients, neurotransmitters are altered, downward inhibitory tone is reduced. Do chronic pain patients exhibit cognitive deficits?
Line 174 – reference needed.
Lines 177-185 – What details related to the compositions of the LED and VLED diets are provided? Did they meet minimum nutritional recommendations? How substantial was the reported pain reduction for patients enrolled in this trial?
Line 186 – Are low FODMAP diets truly calorie-restricted or are the calories made up in another class, such as fats or protein?
Lines 200-204 – Discuss the personalized diet consultations (e.g., increase intake of nutrient-dense foods, what kinds, etc). Why cherry juice vs apple juice (e.g., high vs low anthocyanin content)?
Line 204-233 – Please discuss the details of the diet therapy or diet personalization. Without emphasizing the diet details, what is the point of including them in the review at all? This is especially true for references 27, 31, and 32 in which there is not even mention of diet intervention.
Line 235 – What does high-fat diet mean here? High saturated, mono-, and/or polyunsaturated fats? Does a diet enriched in omega-3 lipids constitute a high-fat diet too? I would consider moving the omega-3-focused studies (references 75, 76) here instead of placing them under antioxidant vitamins and minerals.
Line 236: Arachidonic acid and eicosapentanoic acid are not essential (meaning that they must be supplied through diet) dietary fatty acids as they both can be synthesized in humans. Linoleic acid (omega-6) and alpha-linolenic acid (omega-3) however are essential and must come exclusively from diet. Tryptophan is an amino acid, not a lipid. Also, arachidonic acid is the precursor to pronociceptive eicosanoids (prostaglandins, thromboxanes, leukotrienes, etc) and they affect both the peripheral and central nervous systems.
Line 240 – reference needed.
Line 241 – reference needed.
Lines 250-259 – Details of diet are needed. What is the source of dietary lipids? Of what lipid class?
Lines 306-313 – Details of diet are needed. Is it also low-fat diet? Plant-based doesn’t necessarily equate to low-fat as avocadoes, olives, nuts, seeds are high-fat plant foods commonly consumed.
Lines 361-371 – This section could be removed as the DASH trial was not investigating the role of sodium content in patients with chronic headache, but rather normal everyday headache risk (i.e., not chronic pain).
Line 336 - reference needed. What types of chronic pain also present with comorbid muscle atrophy?
Line 344 – What was the control condition?
Line 347 – Details needed pertaining to the type, amount of protein intake.
Line 388 – What was the clinical improvement observed? Was it pain? Was the elimination diet the same for all patients or different depending on specific food allergies?
Line 391 – For MSG elimination diets, did these preclude consumption of foods known to contain various amounts of MSG, such as tomatoes and cheeses? What about substitutes for aspartame? Could any of the results from the trials be the result of diet non-adherence by enrolled subjects?
Lines 421-439 – Did these gluten-free diets still provide full nutritional supplementation? There was mention of this on Line 440. It is well-known, for example, that B12 deficiency can manifest as peripheral neuropathy. Was there any information pertaining to reduction in inflammatory biomarkers?
Lines 459-460 – The reduction in abdominal pain in subjects without the mutation is quite intriguing. What additional information is available concerning the specific diet composition or nutritional details aside from being lactose-free? Any information warrants mention here or in the discussion.
Lines 466-490 – Histamine-free diet section was nicely written, however there is no mention of specific trials testing such a diet on chronic pain conditions. Yet, Line 485-487 mentions that the most studied conditions are headache, abdominal, musculoskeletal, and neuropathic pain. Where are those studies and why are they not discussed in detail? The same goes for Line 487-489, if the standard of care for vascular headaches is a histamine-free diet, where is the trial information?
Line 497 – Similar to the DASH trial, did the SU.VI.MAX trial evaluate vitamins and minerals in the context of chronic pain? They state specifically that patients enrolled at beginning were devoid of chronic diseases. Function-limiting pain certainly could include some with chronic pain, but also those without.
Line 515-516 – Change “statistic signification” to “statistical significance”. In regards to the appositive, what measurable outcome(s) was not significant in this trial, yet warranted such a bold conclusion by the authors?
Lines 529-537 – Omega-3 fatty acid studies should be placed in a different category as their effects are anti-inflammatory and anti-nociceptive through mechanisms beyond being anti-oxidative. Also are the mechanisms known how omega-6 fatty acids are pronociceptive whereas omega-3 fatty acids are anti-nociceptive? There is significant research on these mechanisms and perhaps worth including.
Line 532 – I would remove “results were as expected” because there was nothing expected about enriched omega-3 lipids via diet improving migraine-associated pain.
Line 554-555 – There was no change in inflammatory biomarkers in the blueberry group vs placebo, yet improved WOMAC scores. Does this speak to the antioxidant effects of blueberries then per the authors?
Line 557 – Reference 25 was previously mentioned under the Lifestyle section. To avoid confusion, I would bring it up only in one place. I think it works best under fruits and fibers rather than lifestyle.
Line 576 – Reference 80, again was this study investigating a chronic pain condition or rather just daily pain in older adults? Consider removing.
Line 585 – Did these Parkinson’s patients have chronic abdominal pain as defined in the introduction?
Line 603 – Is NKCP made with or from soy? Assuming so, are there other compositional considerations that might influence pain other than the probiotic (e.g., soy is loaded with polyunsaturated fatty acids and B vitamins)?
Line 605 – What was the nature of the enrolled subjects’ shoulder and neck stiffness that made it a chronic pain condition? Arthritis? Surgery?
Line 624 – Far more patient-reported pain indices were utilized in these studies than solely VAS scores. Did any studies utilize physician-assessed indices or objective laboratory measurements?
Line 629-632 – Shouldn’t the number of trials equal 46, the number of papers investigated? Where is abdominal pain? IBS? How are these conditions categorized by the European Pain Federation?
Line 634-644 – Is inflammation and vitamin imbalance the only factors involved in fibromyalgia? Do all fibromyalgia patients respond equitably to diet? Like most chronic pain conditions, fibromyalgia patients are largely heterogeneous, such that the etiologies can differ from patient to patient. One could argue marginal impact of diet on autoimmune-related fibromyalgia. Thus, it is important to highlight here and for all chronic pain conditions that while diet may be beneficial in many cases, it unlikely will help all.
Line 664 – Change “overweight” to “obesity”
Lines 684-686 – Chronic pain is prevalent in patients with rheumatoid arthritis, ulcerative colitis, etc mentioned here. Why are these studies not incorporated into the review as well?
Lines 684-704 – The authors hit on a particularly important point here regarding the therapeutic influence of elevated omega-3 lipids to omega-6 lipids. This is the type of specific details needed to be emphasized when at all possible. The current imbalance of omega-6/omega-3 in the westernized diet is staggering and is thought to “prime” individuals into a pro-inflammatory state. And although people persistently on a Western-style diet may not say they are constantly in pain, more evidence is emerging that this “primed state” exacerbates co-morbid pain conditions, including those that become chronic. More research such as this is absolutely needed. Very nice point.
Author Response
Reviewer 3
Thank you for your observations. According to your suggestions, we have made following corrections:
- We have rewritten the introduction adding more observations on diet and nutrients integrating the types of pain in the context of diets, dietary constituents and supplements. We have provided a background of current knowledge in the field.
- We have proof-read the whole article and we have made all necessary corrections of English language
- We have realigned the figure and the table text.
- We have remade Table 1 entirely. We have re-labeled “Conditions/Comorbidities” to “Chronic Pain Condition.” We have reviewed once more the studies included to validate the chronic pain context. We have added another column to “Outcomes” to indicate if the chronic pain condition in the specific study was a primary or secondary clinical outcome. We have regrouped the studies in 8 dietary intervention categories: Caloric restriction and fasting, Enriched polyunsaturated fatty acid diets, Low-fat plant-based diets, High-protein diets, Elimination diets, Antioxidant vitamins and minerals, Fruits and fibers and Prebiotics and probiotics in the table and in the text. For each study, we have clearly specified the pain condition and comorbidities in the respective rows.
- By request of all reviewers we have eliminated the DASH trial (low sodium) because it investigated daily headache risk in otherwise healthy subjects, which is not chronic pain. Also the SU.VI.MAX trial which evaluated the effect of dietary vitamins and minerals on function-limiting pain in older adults, which is not explicitly chronic pain, was excluded from our review. We have entirely removed the Personalized Lifestyle and Healthy Lifestyle” section and have incorporated trials related to diet composition into other sections. We checked again the trials previously included in this section, and we removed the ones that made no reference to diet composition. Omega-3 fatty acid studies were placed under Enriched polyunsaturated fatty acid diets, instead of being kept under Antioxidant vitamins and minerals.
- We have limited our search to PubMed because the new up-graded version has offered the best filters for selection, as stated in the search strategy, allowing an accurate and rapid search according to our aim. It also is the most comprehensive medical database. Filters on Web of Science include “article”, “review”, “meeting abstract”, “proceedings paper” and “editorial material”, which require further time-consuming selections and filtering. Custom year range allows only searches until 2018. In Scopus the combined search “chronic pain” and “diet” were split to “chronic”, “pain” and “diet”, resulting in faulty redundant selections. This was an invited review with a very tight dead-line. We selected the database that seemed best to fit our purpose.
- The decision to limit our searches to the last decade was also made because the “Advanced search” section in PubMed offers a graph of numbers of publications in the selected field per year, and more than 80% of the articles fulfilling the criteria to be analyzed in our review were published in the last 10 years, the rest being scattered throughout the previous 40 decades and implying a sorting effort for which we did not have enough time.
- We have included the study limitations and have rewritten the conclusions

Round 2
Reviewer 1 Report
The authors have now substantially improved the manuscript in this revision. I also prefer the format and presentation of revised Table 1. One minor comment is that the authors may consider splitting this table by different pain types which may make it even easier for the prospective readers to browse through the relevant data included in this review.
Author Response
Thank you for your suggestions. We have added another table referring to chronic pain categories and dietary pattern and interventions in the sub-chapter of Discussions, for an easy-to-read format. We have also proof-checked grammar and spelling errors.

Reviewer 3 Report
Major Issue:
Writing a review on a topic of this magnitude is extremely difficult. I again commend the authors for taking on this task, because it is an important topic that needs to be comprehensively evaluated in order to help direct the next phases of preclinical and clinical research. This resubmission is much improved from the original. However, due to significant omissions in the clinical pain literature, additional work still is needed in order for this manuscript to achieve its stated objective.
Performing a thorough literature search on any topic, let alone this topic, is time-consuming and tedious. However, those difficulties are not adequate justification for using a severely constrained search approach for the sake of time. I also strongly disagree with the notion that “a very tight deadline” justifies limiting the literature search and rushing to publish. We the reviewers were not made aware that this was an invited review (nor should we have been). We also have no idea how much time the authors were given by Nutrients to prepare this manuscript. Considering the scope here, any expectation of the authors to prepare this manuscript on an extremely short timeline would seem very odd, and frankly unreasonable.
What needs to be completely revamped is the search strategy. In its current form, the search strategy is missing far too many relevant clinical studies related to dietary interventions for various chronic pain conditions. We all agree that PubMed is an amazing repository of biomedical research literature. Because of its enormity however, searching for topic-relevant papers and sifting through search results is difficult. It is an unrealistic expectation that a single keyword/filter-based search will comprehensively, or even adequately, identify all relevant clinical studies on diet and chronic pain.
What could improve the search parameters is consideration of what chronic pain represents (partly covered in the manuscript introduction). In the pain field, which includes researchers, clinicians, clinician-scientists, nurses, etc, there remains an ongoing debate on how to best define “chronic pain.” Historically, the search for a generally-accepted definition of clinical chronic pain has revolved around identifying a single bonafide (albeit arbitrary) timeline that describes the initiation, but also the persistence of patient pain symptoms. Unfortunately, there is no 1 definition that can sufficiently encompass all the different types of chronic pain, the heterogeneity of symptoms, and the heterogeneity of the patient populations. Because of this, the use of “chronic” as the primary identifier of a clinical pain condition has been waning for some time. Therefore, it should not be surprising that relevant studies on many important chronic pain conditions are missed entirely because they prioritize designations other than “chronic” or “chronic pain.” It is much more common to classify based on the diagnosis, the etiology, or the clinical symptomatology, for example. This is true for some of the most prevalent chronic pain conditions in the past 10-20 years worldwide, including, but not limited to, chemotherapy-induced peripheral neuropathy, diabetic neuropathy, HIV-induced neuropathy, burn injury-induced pain, multiple sclerosis, spinal cord injury-induced pain, trigeminal neuralgia, and temporomandibular disorder. To illustrate this point, performing a specific search such as clinical trials testing “vitamin E supplementation” or “antioxidant supplementation” in patients with chemotherapy-induced peripheral neuropathy yields published results from many clinical trials conducted in the past 10 years. Unfortunately, none of these studies have been identified under “Section 3.6 Antioxidant Vitamins and Minerals”. Moreover, most of the aforementioned pain conditions, despite representing an enormous cross-section of patients that deal with chronic pain every single day, are not even mentioned. A more robust search strategy will show that many studies have been published on investigating the therapeutic impact of various dietary interventions. Excluding them on the basis of the limited search strategy not only diminishes the impact of this review, but it also misrepresents the research progress that has been done up to this point.
In consideration of these points, the literature search strategy must be changed at the minimum. Perhaps it is more reasonable to constrain the scope of the review. Alternatively, a more focused study inclusion criteria could be used to “define” the most impactful studies from the past decade (e.g., cutoff for minimum number of participants, RCTs only)? Whatever the decision, the clinical literature analysis must be improved if to adequately inform on the dietary interventions used for chronic pain.
Specific Comments:
*Note: I stopped providing detailed comments in consideration of the major issue stated above.
Line 92-95 – Might add that western diet imbalance also yields fewer anti-inflammatory mediators, including antioxidants and anti-oxidant defense. The composition of a western-style diet may not necessarily increase inflammation directly, but rather induce a reduction in anti-inflammatory defense.
Line 93-94 – Some slight redundancy: interleukins, TNF-alpha are cytokines
Line 94 – Consider changing “nociceptors” to “peripheral afferent neurons” or “peripheral neurons,” as the pro-inflammatory milieu can certainly sensitize nociceptors and non-nociceptors alike that collectively contribute to a chronic pain state.
Line 94 – “including”
Line 96 – I would be careful the way you bring up spontaneous pain and hyperalgesia here. Might be best just as “chronic pain often results from a persistent pro-inflammatory state.” Emphasizing the temporal component of the underlying mechanisms is actually critical, since acute or transient inflammation generally resolves quickly without risk for prolonged impact.
Line 100-101 – Remove “plant-based or Mediterranean diets” since they are not the only diets that can lead to reduced inflammation. What about traditional Eastern diets (e.g., Japanese, Pacific Islander, etc)?
Line 140 – What kind of studies? Preclinical? Clinical? If clinical, they should show up in the studies reviewed for this manuscript, correct? Diabetic neuropathy, rheumatoid arthritis, and pancreatitis all produce chronic pain.
Line 154 – Change to “unclear” or “ambiguous”, instead of “elusive?”
Line 156 – Combine with previous paragraph.
Line 228-231 – Check diet categories and mention them in the same order as they are in the first column of Table 1.
Line 248 – Table 1 is much improved. Further improvements could include consistency in certain details and more information in specific boxes.
- Under Type of Study, more information can be provided than just “clinical trial.” Considering that all these studies have been published in the past 10 years, most if not all should conform to contemporary clinical trial design. No study should just be labeled “clinical trial.” If a study is not some form of RCT, pilot interventional study, or privately-funded study, and only satisfies a label like “clinical study,” then perhaps an asterisk is warranted that would include at the end of the table an explanation of the study specifics, and why it was even included (e.g., no control group). Why are some studies RCT and others are randomized? Recommend changing RCT to randomized with other details (even if deduced).
- Under Number of Patients, include total number of participants, then in parentheses the number of males vs females. Perhaps include some useful demographic information here too, such as participant age range and/or ethnicity.
- Under Chronic Pain Condition, add primary or secondary outcome in parentheses after condition and remove the final column instead.
- Under Diet Intervention, some studies have great descriptions of the intervention being tested whereas some could be much improved. For example, diet specific information for Ramsden et al., 2017, which were published in an associated article, should replace linoleic acid lowering diet.
- Under Effects on Pain, more specific details would much improve the current format for certain studies. For example, with Holton et al., 2012, instead of just “improvement in pain symptoms,” concisely, but specifically describe the improvement(s) observed.
Author Response
Thank you for your suggestions. We have remade our search and added as search keys all chronic pain conditions mentioned in the introduction i.e. chronic musculoskeletal pain, chronic neuropathic pain etc and diet, dietary intervention and dietary supplements and the search provided 2 more studies corresponding to our requirements that we have included in the review. Both refer to vitamin E. Many other chronic pain conditions influenced by diets or dietary supplements are evaluated by specific questionnaires like Fatigue Severity Scale (FSS) and Modified Fatigue Impact Scale (MFIS)], in relapsing-remitting MS (RRMS) in multiple sclerosis or Bath Ankylosing Spondylitis Disease Activity Index (BASDAI) ≥ 3, Bath Ankylosing Spondylitis Functional Index (BASFI) ≥ 3, Maastricht Ankylosing Spondylitis Enthesitis Score (MASES) ≥ 2, or peripheral joint count ≥ 2 in active spondyloarthritis, or Disease activity score-28 (DAS28), European League Against Rheumatism (EULAR) response and global health (GH) score in rheumatoid arthritis, without offering concrete data about the evolution of chronic pain, and we have not included them in our review. We have added more information and comments in discussions regarding controversies on certain diets or supplements and research on recent findings explaining the effects of some interventions on nociception.
We have made all required changes in the lines below, as suggested.
Line 92-95 – Might add that western diet imbalance also yields fewer anti-inflammatory mediators, including antioxidants and anti-oxidant defense. The composition of a western-style diet may not necessarily increase inflammation directly, but rather induce a reduction in anti-inflammatory defense.
Line 93-94 – Some slight redundancy: interleukins, TNF-alpha are cytokines
Line 94 – Consider changing “nociceptors” to “peripheral afferent neurons” or “peripheral neurons,” as the pro-inflammatory milieu can certainly sensitize nociceptors and non-nociceptors alike that collectively contribute to a chronic pain state.
Line 94 – “including”
Line 96 – I would be careful the way you bring up spontaneous pain and hyperalgesia here. Might be best just as “chronic pain often results from a persistent pro-inflammatory state.” Emphasizing the temporal component of the underlying mechanisms is actually critical, since acute or transient inflammation generally resolves quickly without risk for prolonged impact.
Line 100-101 – Remove “plant-based or Mediterranean diets” since they are not the only diets that can lead to reduced inflammation. What about traditional Eastern diets (e.g., Japanese, Pacific Islander, etc)?
Line 140 – What kind of studies? Preclinical? Clinical? If clinical, they should show up in the studies reviewed for this manuscript, correct? Diabetic neuropathy, rheumatoid arthritis, and pancreatitis all produce chronic pain. reviews
Line 154 – Change to “unclear” or “ambiguous”, instead of “elusive?”
Line 156 – Combine with previous paragraph.
Line 228-231 – Check diet categories and mention them in the same order as they are in the first column of Table 1.
We have changed Table 1 introducing the information in the specific boxes as you suggested, respecting the indications below:
- Under Type of Study, more information can be provided than just “clinical trial.” Considering that all these studies have been published in the past 10 years, most if not all should conform to contemporary clinical trial design. No study should just be labeled “clinical trial.” If a study is not some form of RCT, pilot interventional study, or privately-funded study, and only satisfies a label like “clinical study,” then perhaps an asterisk is warranted that would include at the end of the table an explanation of the study specifics, and why it was even included (e.g., no control group). Why are some studies RCT and others are randomized? Recommend changing RCT to randomized with other details (even if deduced).
- Under Number of Patients, include total number of participants, then in parentheses the number of males vs females. Perhaps include some useful demographic information here too, such as participant age range and/or ethnicity.
- Under Chronic Pain Condition, add primary or secondary outcome in parentheses after condition and remove the final column instead.
- Under Diet Intervention, some studies have great descriptions of the intervention being tested whereas some could be much improved. For example, diet specific information for Ramsden et al., 2017, which were published in an associated article, should replace linoleic acid lowering diet.
- Under Effects on Pain, more specific details would much improve the current format for certain studies. For example, with Holton et al., 2012, instead of just “improvement in pain symptoms,” concisely, but specifically describe the improvement(s) observed.
According to suggestions of reviewer 1 we have added another table in discussions to allow a faster orientation for the reader on dietary interventions in specific chronic pain conditions.
